# SARS-CoV-2 N protein-induced Dicer, XPO5, SRSF3, and hnRNPA3 downregulation causes pneumonia

Yu-Wei Luo [1,10], Jiang-Peng Zhou[1,10], Hongyu Ji[1,10], Doudou Xu [2,10], Anqi Zheng[3], Xin Wang[1], Zhizheng Dai[1], Zhicheng Luo[1,3], Fang Cao[1], Xing-Yue Wang[3], Yunfang Bai[1], Di Chen[1], Yueming Chen[3], Qi Wang[4], Yaying Yang[5], Xinghai Zhang[6], Sandra Chiu[7,8], Xiaozhong Peng [2,9] ✉, Ai-Long Huang [1] ✉ & Kai-Fu Tang [1] ✉

Though RNAi and RNA-splicing machineries are involved in regulating severe acute respiratory syndrome coronavirus 2 (SARS-CoV-2) replication, their precise roles in coronavirus disease 2019 (COVID-19) pathogenesis remain unclear. Herein, we show that decreased RNAi component (Dicer and XPO5) and splicing factor (SRSF3 and hnRNPA3) expression correlate with increased COVID-19 severity. SARS-CoV-2 N protein induces the autophagic degradation of Dicer, XPO5, SRSF3, and hnRNPA3, inhibiting miRNA biogenesis and RNA splicing and triggering DNA damage, proteotoxic stress, and pneumonia. Dicer, XPO5, SRSF3, and hnRNPA3 knockdown increases, while their over-expression decreases, N protein-induced pneumonia's severity. Older mice show lower expression of Dicer, XPO5, SRSF3, and hnRNPA3 in their lung tissues and exhibit more severe N protein-induced pneumonia than younger mice. PJ34, a poly(ADP-ribose) polymerase inhibitor, or anastrozole, an aromatase inhibitor, ameliorates N protein- or SARS-CoV-2-induced pneumonia by restoring Dicer, XPO5, SRSF3, and hnRNPA3 expression. These findings will aid in developing improved treatments for SARS-CoV-2-associated pneumonia.

Coronavirus disease 2019 (COVID-19), caused by severe acute respiratory syndrome coronavirus 2 (SARS-CoV-2), is highly heterogeneous, ranging from asymptomatic and mild to severe and fatal[1]. Although COVID-19 indiscriminately affects individuals of all age groups, severe pneumonia and mortality occur disproportionately in older individuals[2–5]. An age-dependent increase in disease severity was also observed in a mouse model of COVID-19[6]. However, the basis for age-related differences in the severity of COVID-19 is poorly understood.

[1]Key Laboratory of Molecular Biology on Infectious Disease, Ministry of Education, Chongqing Medical University, Chongqing, PR China. [2]State Key Laboratory of Respiratory Health and Multimorbidity, Key Laboratory of Pathogen Infection Prevention and Control (Peking Union Medical College), Ministry of Education, National Center of Technology Innovation for animal model, CAMS & PUMC, Beijing, PR China. [3]Key Laboratory of Diagnosis and Treatment of Severe Hepato-Pancreatic Diseases of Zhejiang Province, The First Affiliated Hospital of Wenzhou Medical University, Wenzhou, Zhejiang, PR China. [4]Department of Basic Medicine, Chongqing Medical University, Chongqing, PR China. [5]Department of Pathology, Molecular Medicine and Cancer Research Center, Molecular Medicine Diagnostic and Testing Center, Chongqing Medical University, Chongqing, PR China. [6]State Key Laboratory of Virology, Wuhan Institute of Virology, Center for Biosafety Mega-Science, Chinese Academy of Sciences, Wuhan, Hubei, PR China. [7]Division of Life Sciences and Medicine, University of Science and Technology of China, Hefei, Anhui, PR China. [8]Key Laboratory of Anhui Province for Emerging and Reemerging Infectious Diseases, Hefei, Anhui, PR China. [9]Institute of Basic Medical Sciences, Chinese Academy of Medical Sciences, School of Basic Medicine Peking Union Medical College, Beijing, PR China. [10]These authors contributed equally: Yu-Wei Luo, Jiang-Peng Zhou, Hongyu Ji, Doudou Xu. ✉e-mail: pengxiaozhong@pumc.edu.cn; ahuang@cqmu.edu.cn; tangkaifu@cqmu.edu.cn

DNA damage is a pivotal contributor to aging, affecting most aspects of the aging phenotype[7–9]. It is also a driver of COVID-19 pathogenesis. Specifically, the DNA damage response facilitates SARS-CoV-2 infection by boosting the transcription of angiotensin-converting enzyme 2 (ACE2), the major cell entry receptor for SARS-CoV-2[10,11]. Additionally, it may promote SARS-CoV-2 replication, as inhibiting the DNA damage response inhibits SARS-CoV-2 replication[12]. DNA damage may increase COVID-19 severity by inducing cellular senescence, which contributes to the "cytokine storm" in COVID-19 patients[3,13]. DNA damage also causes cytosolic DNA accumulation, which stimulates proinflammatory cytokine production[7,14,15]. Finally, DNA damage induces natural killer group 2, member D (NKG2D) ligand expression. Receptor engagement by NKG2D ligands triggers cytolysis and the release of proinflammatory factors by natural killer and other immune cells[16,17].

Loss of proteostasis, another hallmark of aging, causes unrestrained inflammasome activation, accumulation of dysfunctional organelles, and cellular damage, and is associated with several age-related morbidities[9,18,19]. Deregulation of proteostasis may contribute to COVID-19 pathogenesis as SARS-CoV-2 infection disturbs cellular proteostasis, and proteostasis perturbation inhibits SARS-CoV-2 replication[20,21].

Dicer, a key component of the RNA interference (RNAi) pathway, is downregulated in human and mouse adipose tissues and in *Caenorhabditis elegans* during aging[22]. Age-related deregulation of RNAi components might be associated with hallmarks of aging, as the knockdown of RNAi components leads to DNA damage and probably affects proteostasis by increasing protein synthesis owing to global downregulation of microRNAs (miRNAs)[23–29]. Additionally, the RNAi pathway may inhibit SARS-CoV-2 replication via RNA degradation. Mechanistically, Dicer cleaves viral double-stranded RNAs (dsRNAs) to generate small interfering RNAs (siRNAs), which guide the sequence-specific degradation of viral RNAs[30,31]. As a countermeasure, viruses encode suppressors to antagonize RNAi by sequestering Dicer and/or dsRNAs[32,33]. However, the detailed role of RNAi components in COVID-19 pathogenesis requires further research.

Age-related deregulation of splicing factors occurs in multiple tissues (blood, brain, muscle, skin, and liver) of different organisms (mice, rats, and humans) and is associated with genomic stability and proteostasis[34–38]. Knockdown of splicing factors leads to the accumulation of R-loop, a transcription-coupled DNA-RNA hybrid that can induce DNA damage[35–37]. Moreover, inhibition of RNA splicing leads to the production of intron-containing truncated proteins, a subset of which have intrinsically disordered regions, form insoluble cellular condensates, and trigger a proteotoxic stress response[38]. RNA splicing is essential for SARS-CoV-2 replication, as splicing inhibition results in impeded viral replication[20]. However, the precise role of RNA-splicing factors in COVID-19 pathogenesis requires further clarification.

Here, we show that SARS-CoV-2 N protein induces the autophagic degradation of two RNAi components (Dicer and XPO5) and two splicing factors (SRSF3 and hnRNPA3), thereby inhibiting miRNA biogenesis and RNA splicing and triggering DNA damage, proteotoxic stress, and pneumonia. Age-associated downregulation of Dicer, XPO5, SRSF3, and hnRNPA3 expression in lung tissues increases the severity of N protein-induced pneumonia. Treatment with PJ34, a poly (ADP-ribose) polymerase inhibitor, or anastrozole, an aromatase inhibitor, alleviates the N protein-induced pneumonia. This study may provide new insights into the pathogenesis of SARS-CoV-2-induced pneumonia, potentially helping in the development of improved prevention and treatment strategies.

## Results

### RNAi components and splicing factors are downregulated in patients with severe COVID-19

To investigate whether the expression levels of RNAi components and splicing factors are associated with COVID-19 severity, we analyzed the published RNA sequencing data of monocytic myeloid-derived suppressor cells (M-MDSC) from patients with severe or asymptomatic COVID-19[39]. The mRNA levels of the RNAi components and splicing factors in M-MDSCs were lower in patients with severe COVID-19 than in those with asymptomatic COVID-19 (Supplementary Fig. 1a). By analyzing another set of published RNA sequencing data[40], we found that the levels of RNAi components and splicing factors were also lower in the lung tissues from deceased COVID-19 patients than in those from individuals without COVID-19 (Supplementary Fig. 1b). Collectively, these results suggest that decreased expression of RNAi components and splicing factors is associated with increased COVID-19 severity.

### SARS-CoV-2 nucleocapsid (N) protein interacts with and induces autophagic degradation of Dicer, XPO5, SRSF3, and hnRNPA3

To investigate whether SARS-CoV-2 modulates the function of the RNAi pathway, a cellular reversal-of-silencing assay was performed[33]. SARS-CoV-2 suppressed RNAi (Fig. 1a and Supplementary Fig. 1c). Using an intron-containing luciferase reporter minigene[41], we demonstrated that SARS-CoV-2 inhibited the splicing of this minigene (Fig. 1b).

Next, to investigate whether SARS-CoV-2-encoded proteins are responsible for the inhibitory effect on RNAi and RNA splicing, four plasmids expressing the structural proteins encoded by SARS-CoV-2 were co-transfected with reporter plasmids. We found that N protein repressed both RNAi and RNA splicing (Fig. 1c, d and Supplementary Fig. 1d).

To understand the molecular mechanisms underlying the N protein-mediated regulation of RNAi and RNA splicing, putative protein interactors of N protein were retrieved from the Biological General Repository for Interaction Datasets (BioGRID)[42]. We observed that the RNAi components and splicing factors may interact with N protein (Supplementary Fig. 1e). Co-immunoprecipitation assays confirmed the interaction between N protein and Dicer, exportin 5 (XPO5), serine/arginine-rich splicing factor 3 (SRSF3), and heterogeneous nuclear ribonucleoprotein A3 (hnRNPA3) in human lung cancer (A549) and human normal lung epithelial (BEAS-2B) cell lines ectopically expressing N protein (Fig. 1e and Supplementary Fig. 2a). Interactions between N protein and Dicer, XPO5, SRSF3, and hnRNPA3 were also detected in SARS-CoV-2-infected A549-hACE2 cells (Fig. 1f and Supplementary Fig. 2b). Consistent with the reports that N protein is an RNA-binding protein[43–45], RNA-immunoprecipitation assay revealed its association with pre-miRNA (Fig. 1g). RNase treatment disrupted the interaction between N protein and Dicer, XPO5, SRSF3, and hnRNPA3 (Fig. 1h), suggesting that these interactions are RNA-dependent.

Subsequently, we investigated whether N protein regulates the expression of Dicer, XPO5, SRSF3, and hnRNPA3. Although N protein did not affect the mRNA levels of *Dicer, XPO5, SRSF3*, or *hnRNPA3* (Supplementary Fig. 2c), it decreased the abundance of their proteins in A549 and BEAS-2B cells (Fig. 1i, and Supplementary Fig. 2d). Moreover, SARS-CoV-2 infection also led to the downregulation of Dicer, XPO5, SRSF3, and hnRNPA3 at the protein level but not at the mRNA level (Fig. 1j, Supplementary Fig. 2e). As controls, the spike, membrane, and envelope proteins had no effect on the expression of Dicer, XPO5, SRSF3, or hnRNPA3 at both protein and mRNA levels (Supplementary Fig. 2f, g). Treatment with autophagy inhibitors chloroquine or bafilomycin A1 relieved the N protein- or SARS-CoV-2-induced downregulation of Dicer, XPO5, SRSF3, and hnRNPA3 (Fig. 1k, l, Supplementary Fig. 2h). However, treatment with a proteasome inhibitor, MG132, did not affect the N protein-induced downregulation of Dicer, XPO5, SRSF3, and hnRNPA3 proteins (Supplementary Fig. 2i), suggesting that the effect of N protein on the expression of Dicer, XPO5, SRSF3, and hnRNPA3 proteins is autophagy-dependent.

N protein was found to induce autophagy (Fig. 2a, b). A search of the BioGRID database indicated that N protein might interact with

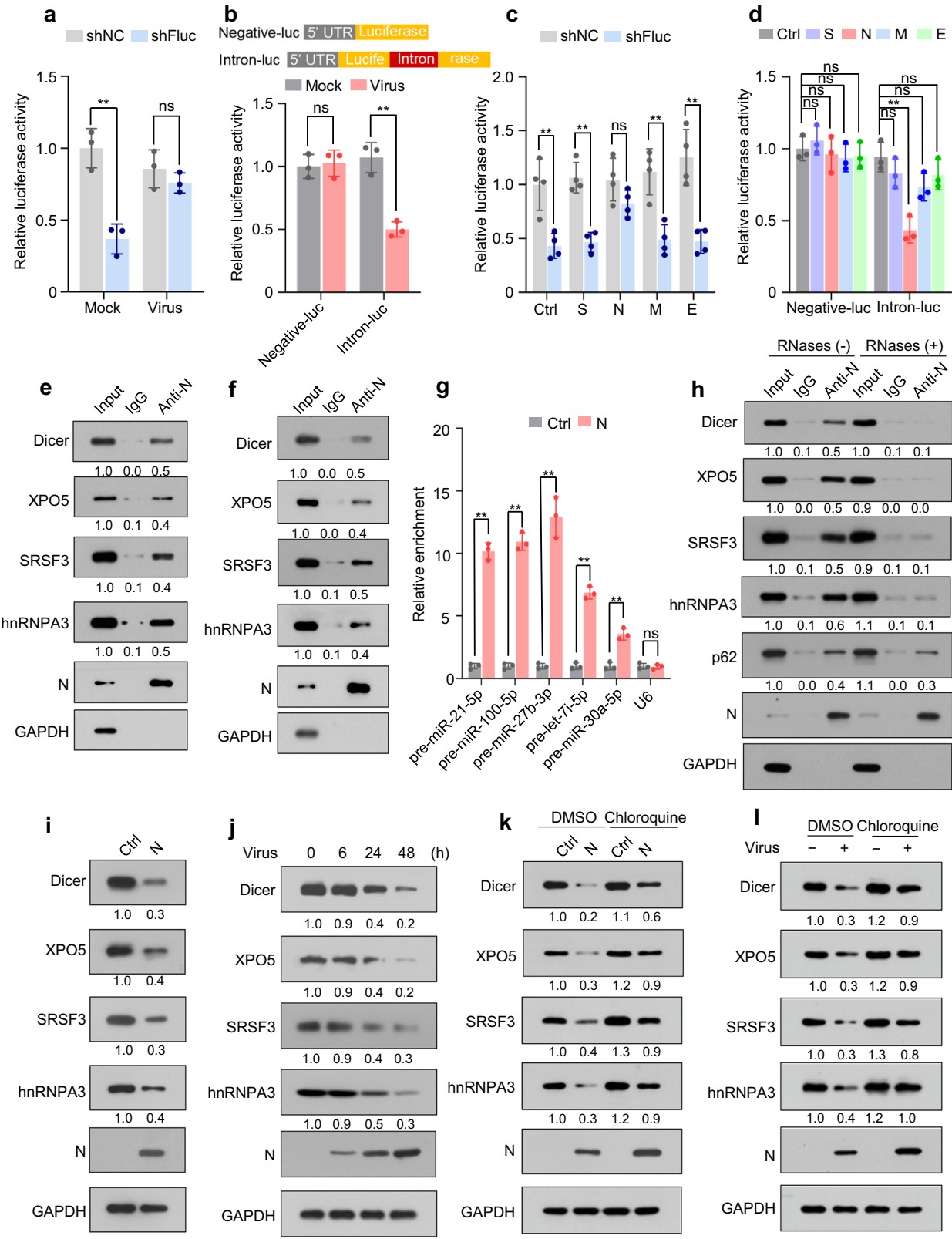

SQSTM1/p62, an autophagy receptor protein recruiting specific cargoes to phagophores[46]. Confocal microscopy analyses and co-immunoprecipitation analyses confirmed the interaction between N protein and p62 and their colocalization, respectively (Fig. 2c–f). RNase treatment did not obviously disrupt the interaction between N protein and p62 (Fig. 1h). Moreover, we found that p62 was associated with Dicer, XPO5, SRSF3, and hnRNPA3 in N protein-expressing cells

and SARS-CoV-2-infected cells but not in cells without N protein expression (Fig. 2e, f). p62 knockdown partially relieved N protein-induced downregulation of Dicer, XPO5, SRSF3, and hnRNPA3 proteins; however, it did not impact the Dicer, XPO5, SRSF3, or hnRNPA3 protein levels in cells not expressing N protein (Fig. 2g).

Collectively, these findings indicate that p62 recruits the N protein-interacting proteins, including Dicer, XPO5, SRSF3, and

**Fig. 1 | SARS-CoV-2 N protein represses Dicer, XPO5, SRSF3, and hnRNPA3 expression and inhibits RNAi and RNA splicing.** See also Supplementary Figs. 1, 2. **a** HEK293T-hACE2 cells were co-transfected with plasmids encoding firefly luciferase (pGL3-Control) and its shRNA (shFluc) or control shRNA (shNC) and then infected with SARS-CoV-2 or mock. Luciferase assay was performed 24 h post-infection. **b** Upper: schematic of RNA-splicing reporters; lower: HEK293T-hACE2 cells were transfected with the splicing reporters and infected with SARS-CoV-2 or mock. Luciferase assays were performed 24 h post-infection. **c** HEK293T cells were co-transfected with pGL3-Control, shFluc or shNC, and plasmids expressing different strep-tagged viral proteins. Luciferase assay was performed 48 h post-transfection. **d** Splicing reporters were co-transfected with plasmids encoding various viral proteins into HEK293T cells. Luciferase assay was performed 48 h post-transfection. **e**, **f** Lysates of A549 cells stably expressing N protein (A549-N; **e**) or SARS-CoV-2-infected A549-hACE2 cells (**f**) were subjected to immunoprecipitation using anti-N protein antibody and immunoblotting with the indicated antibodies. **g** Lysates of A549-N cells were subjected to RNA-immunoprecipitation using an

anti-N protein antibody, followed by quantification of the indicated pre-miRNAs and *U6*. **h** Lysates of A549-N cells were immunoprecipitated using an anti-N protein antibody in the presence or absence of RNases, followed by immunoblotting with the indicated antibodies. **i** Abundance of the indicated proteins in A549-Ctrl (A549 cells transfected with control plasmid) and A549-N cells. **j** Expression of the indicated proteins in SARS-CoV-2- or mock-infected A549-hACE2 cells was detected at different time points after infection. **k** Abundance of the indicated proteins in A549-Ctrl and A549-N cells treated with or without chloroquine. **l** Abundance of the indicated proteins in SARS-CoV-2- or mock-infected A549-hACE2 cells treated with or without chloroquine. Data in (**a–d**, **g**) are expressed as mean ± standard deviation (SD) of three (**a**, **b**, **d**, **g**) or four (**c**) biological replicates. **$p < 0.01$; ns not significant ($p > 0.05$; two-tailed unpaired Student's $t$-test). The numbers below the blots indicate the relative densitometric quantification of bands normalized to the input (**e**, **f**, **h**) or GAPDH bands (**i–l**); the mean values of three independent experiments are shown. Source data and exact $p$ values are provided in the Source Data file.

hnRNPA3, to phagophores via interactions with N protein; this leads to their autophagic degradation.

## SARS-CoV-2 N protein induces DNA damage by downregulating Dicer, XPO5, SRSF3, and hnRNPA3 expression

As the RNAi machinery and splicing factors are essential for genomic stability[23,24,35–37], we next investigated whether SARS-CoV-2 N protein-induced DNA damage by downregulating Dicer, XPO5, SRSF3, and hnRNPA3 expression. Ectopic N protein expression or SARS-CoV-2 infection induced DNA damage, as indicated by the phosphorylation of ATM, ATR, Chk1, Chk2, and H2AX, DNA break accumulation, and phosphorylated H2AX (γ-H2AX) foci formation (Fig. 3a–e and Supplementary Fig. 3a–c). As controls, membrane or envelope protein ectopic expression did not induce DNA damage (Supplementary Fig. 3d, e). Knockdown of Dicer, XPO5, SRSF3, or hnRNPA3 led to DNA damage (Fig. 3f, g and Supplementary Fig. 4a), whereas their overexpression partially alleviated N protein-induced DNA damage (Fig. 3h and Supplementary Fig. 4b–e).

Defects in the splicing factors SRSF3 and hnRNPA3 increase R-loop formation, resulting in DNA damage[35–37]. We found that ectopic N protein expression or SARS-CoV-2 infection induced R-loop accumulation (Fig. 3i, j); overexpression of RNase H1—an endoribonuclease that degrades the RNA portion of R-loop[47]—partially alleviated this effect and decreased DNA damage accumulation (Fig. 3k, and Supplementary Fig. 4b–f). Overall, these findings indicate that N protein-induced downregulation of Dicer, XPO5, SRSF3, and hnRNPA3 expression results in DNA damage.

## SARS-CoV-2 N protein represses miRNA biogenesis

XPO5 and Dicer play pivotal roles in miRNA biogenesis. XPO5 transports the miRNA precursors (pre-miRNAs) from the nucleus to the cytoplasm, where Dicer processes them into mature miRNAs[23,25]. Small RNA deep sequencing revealed that N protein induced global downregulation of miRNA expression (Fig. 4a and Supplementary Fig. 5a). Moreover, the quantification of the top five miRNAs with the highest expression in A549 and BEAS-2B cells, accounting for ~80% of the entire cellular miRNome, revealed that they were all downregulated in N protein-expressing cells and SARS-CoV-2 infected cells (Fig. 4b, c, and Supplementary Fig. 5b).

Biochemical fractionation experiments revealed that ectopic N protein expression or SARS-CoV-2 infection induced nuclear retention of pre-miRNAs, as indicated by an increase in pre-miRNA levels in the nucleus and their decrease in the cytoplasm (Fig. 4d, e, and Supplementary Fig. 5c), which was relieved via XPO5 overexpression (Fig. 4f, g, and Supplementary Fig. 5d). In contrast, N protein expression or SARS-CoV-2 infection did not induce nuclear retention of 18S rRNA or several mRNAs (Fig. 4d, e, and Supplementary Fig. 5c), excluding the possibility that N protein non-specifically inhibits RNA

export from the nucleus. Moreover, ectopic N protein expression or SARS-CoV-2 infection reduced the ratio between mature miRNA and pre-miRNA (Fig. 4h, i, and Supplementary Fig. 5e), which was relieved by Dicer overexpression (Fig. 4j, k, and Supplementary Fig. 5f). Dicer and XPO5 overexpression also rescued miRNA expression in N protein-expressing and SARS-CoV-2-infected cells (Fig. 4l, m, and Supplementary Fig. 5g).

Overall, these findings indicate that N protein blocks the transportation of pre-miRNAs from the nucleus to the cytoplasm by decreasing XPO5 expression and represses the processing of pre-miRNAs into mature miRNAs by downregulating Dicer expression.

## SARS-CoV-2 N protein represses RNA splicing by decreasing SRSF3 and hnRNPA3 expression

Consistent with previous reports that SRSF3 and hnRNPA3 are splicing factors[36,37], SRSF3 or hnRNPA3 knockdown inhibited the splicing of an intron-containing reporter minigene (Fig. 5a), whereas their overexpression partially relieved the inhibitory effect of N protein or SARS-CoV-2 infection on RNA splicing (Fig. 5b, c). Transcriptome analysis revealed that N protein induced a global increase in intron retention (IR; Fig. 5d, e). Additionally, reverse transcription-quantitative polymerase chain reaction (RT-qPCR) confirmed that ectopic N protein expression or SARS-CoV-2 infection reduced the splicing efficiency of a panel of endogenous introns (Fig. 5f, g). The inhibitory effect elicited by N protein and SARS-CoV-2 on RNA splicing was partially alleviated by SRSF3 and hnRNPA3 overexpression (Fig. 5h, i). These results suggest that N protein represses RNA splicing by decreasing SRSF3 and hnRNPA3 expression.

## SARS-CoV-2 N protein induces proteotoxic stress and activates c-Jun N-terminal kinase

Splicing inhibition induces widespread IR and the subsequent translation of intron-retained mRNAs; a proportion of intron-derived polypeptides are condensation-prone and proteotoxic[38]. Metabolic labeling of nascent polypeptides with puromycin, followed by ultracentrifugation, revealed that ectopic N protein expression or SARS-CoV-2 infection induced increased synthesis of insoluble proteins (Fig. 6a, b). Although N protein can condense with RNA through liquid-liquid phase separation[48–50], we found that it was mainly detected in the soluble fraction, suggesting that it did not form insoluble aggregates (Supplementary Fig. 6a). As a control, ectopic expression of SARS-CoV-2 nonstructural protein 8 (NSP8), another RNA-binding protein, did not increase the synthesis of insoluble proteins (Supplementary Fig. 6b). Moreover, we found that the spike and membrane proteins did not affect the synthesis of insoluble proteins, and the envelope protein even repressed the synthesis of insoluble proteins (Supplementary Fig. 6c).

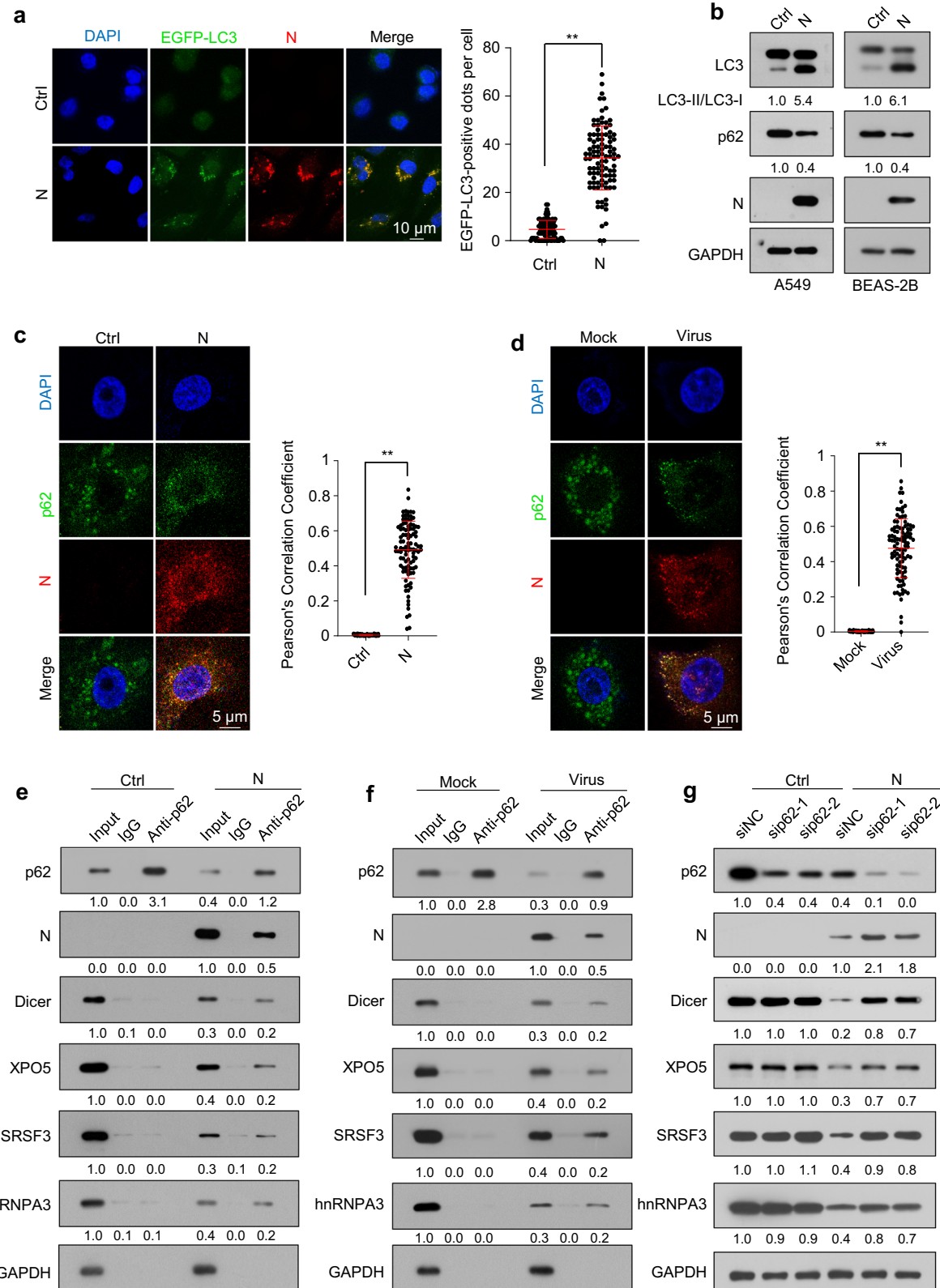

To investigate whether N protein induces proteotoxic stress, we used a firefly luciferase (Fluc) or its conformationally unstable form (R188Q-R261Q double mutant [DM]) fused to an enhanced green fluorescent protein (EGFP), as a reporter. Under proteotoxic stress, FlucDM-EGFP forms aggregates due to chaperone shortage[51]. Concordantly, N protein and SARS-CoV-2 infection induced the production of EGFP aggregates and reduced luciferase activity in cells harboring the FlucDM-EGFP reporter (Fig. 6c–e). As controls, the spike, membrane, and envelope proteins could not induce proteotoxic stress (Supplementary Fig. 6d). Moreover, both N protein ectopic expression and SARS-CoV-2 infection activated c-Jun N-terminal kinase (JNK), a multi-faceted stress-responsive kinase that can be activated by proteotoxic stress[38,52], as evidenced by its increased phosphorylation (Fig. 6f, g). Overall, these findings suggest that N protein induces proteotoxic stress.

**Fig. 2 | SARS-CoV-2 N protein mediates the interactions of Dicer, XPO5, SRSF3, and hnRNPA3 with p62 and induces their autophagic degradation. a** The formation of EGFP-LC3 puncta in A549 cells stably expressing EGFP-LC3 transfected with N protein-expressing plasmid or control plasmid (n = 100 cells from three biological replicates). **b** LC3-II and p62 levels in A549-Ctrl and A549-N cells (left) or BEAS-2B-Ctrl and BEAS-2B-N cells (right). **c** Immunofluorescence of A549-Ctrl or A549-N cells treated with chloroquine using anti-p62 and anti-N protein antibodies and DAPI for nuclear staining. **d** Immunofluorescence of A549-hACE2 cells infected with SARS-CoV-2 or mock-infected and treated with chloroquine using anti-p62 and anti-N protein antibodies and DAPI for nuclear staining. Pearson's correlation coefficient of the colocalization between N protein and p62 in (**c**, **d**) (n = 100 cells from three biological replicates). **e** Cell lysates of A549-Ctrl and A549-N cells were subjected to immunoprecipitation using an anti-p62 antibody and immunoblotting with the indicated antibodies. **f** Cell lysates of A549-hACE2 cells infected with SARS-CoV-2 or mock were subjected to immunoprecipitation using an anti-p62 antibody and immunoblotting with the indicated antibodies. **g** Immunoblotting analysis of the indicated proteins in A549-Ctrl and A549-N cells transfected with a negative control siRNA (siNC) or p62-specific (sip62) siRNAs. The numbers below the blots indicate the relative densitometric quantification of the bands normalized to the corresponding GAPDH band (**b**, **g**) or input (**e**, **f**); the mean values of three independent experiments are shown. Data in (**a**, **c**, **d**) are expressed as mean ± SD. **\*\*p < 0.01 (two-tailed unpaired Student's t-test). Ctrl control plasmid, N N protein, anti-N anti-N protein antibody. Source data and exact p values are provided in the Source Data file.

## SARS-CoV-2 N protein increases protein synthesis

Proteotoxic stress activates JNK, which induces disassembly of the mTORC1 complex, leading to dephosphorylation of eukaryotic translation initiation factor 4E binding protein 1 (4EBP1) and ribosomal protein S6 kinase beta-1 (S6K1) and repressing protein translation[38,52]. Although ectopic N protein expression or SARS-CoV-2 infection led to 4EBP1 dephosphorylation, it increased global protein synthesis without markedly impacting S6K1 phosphorylation (Fig. 6f–h, Supplementary Fig. 6e). As controls, the spike and membrane proteins had no effect on global protein synthesis, while the envelope protein even repressed global protein synthesis (Supplementary Fig. 6f).

Dicer or XPO5 knockdown led to JNK activation and 4EBP1 dephosphorylation while slightly increasing S6K1 phosphorylation and promoting protein synthesis. Simultaneous Dicer and XPO5 knockdown led to a more substantial increase in protein synthesis (Fig. 6i, j). Dicer and XPO5 overexpression partially prevented the N protein-induced increase in protein synthesis and JNK phosphorylation and caused a slight decrease in S6K1 phosphorylation and a slight increase in 4EBP1 phosphorylation in N protein-expressing cells (Fig. 6k, l). SRSF3 or hnRNPA3 knockdown activated JNK, dephosphorylated S6K1 and 4EBP1, and repressed protein synthesis (Fig. 6i, j). Overall, these findings indicate that N protein promotes protein synthesis by decreasing Dicer and XPO5 expression and inhibits protein synthesis by downregulating SRSF3 and hnRNPA3 expression. As the downregulation of Dicer and XPO5 promotes protein synthesis more significantly than the downregulation of SRSF3 and hnRNPA3 inhibits it, N protein ultimately enhances protein synthesis.

## SARS-CoV-2 N protein induces pneumonia in mice

DNA damage and proteotoxic stress can induce cell death[7–9,18,19]. Consistently, N protein induced apoptosis in A549 and BEAS-2B cells (Supplementary Fig. 7a, b). Additionally, DNA damage induces the accumulation of cytosolic DNA, which stimulates the expression of proinflammatory cytokines and NKG2D ligands[7,14–16]. Concordantly, N protein caused cytosolic DNA accumulation and promoted the expression of interferon beta (IFNβ), interleukin-6 (IL-6), and NKG2D ligands (Supplementary Fig. 7c, d).

We also investigated the in vivo effects of N protein and observed that its expression in mouse lung tissues led to the downregulation of Dicer, XPO5, SRSF3, and hnRNPA3, DNA damage, cytosolic DNA accumulation, apoptosis, upregulation of IFNβ, IL-6, and NKG2D ligands, and pneumonia with obvious macrophage infiltration (Fig. 7a–f). As a control, expression of NSP8 in mouse lung tissues did not induce pneumonia (Supplementary Fig. 7e). Given that N protein induced similar levels of lung injury in male and female mice (Fig. 7g), we used only male mice for the following experiments. Dicer, XPO5, SRSF3, and hnRNPA3 knockdown in lung tissues led to lung injury and pneumonia, whereas their overexpression partially alleviated N protein-induced pneumonia (Fig. 7h, i and Supplementary Fig. 7f, g). Furthermore, SARS-CoV-2 infection also led to the downregulation of Dicer, XPO5, SRSF3, and hnRNPA3 in mouse lung tissues (Fig. 7j), and overexpression of Dicer, XPO5, SRSF3, and hnRNPA3 in mouse lung tissues partially alleviated SARS-CoV-2-induced pneumonia (Fig. 7k and Supplementary Fig. 7h). These results suggest that N protein-induced downregulation of Dicer, XPO5, SRSF3, and hnRNPA3 protein expression is a mechanism underlying SARS-CoV-2-induced pneumonia.

## Age-related downregulation of Dicer, XPO5, SRSF3, and hnRNPA3 expression is associated with increased severity of N protein-induced pneumonia

Analyzing the published RNA sequencing data from lung tissues of 3-, 6-, 12-, and 24-month-old C57BL/6J mice revealed that mRNA levels of *Dicer, Xpo5, Srsf3*, and *Hnrnpa3* decreased with age (Fig. 8a). The age-dependent downregulation of Dicer, XPO5, SRSF3, and hnRNPA3 in lung tissues was confirmed at the mRNA and protein levels in 8-week-old and 18-month-old C57BL/6J mice (Fig. 8b, c). Ectopic N protein expression in lung tissues led to DNA damage, apoptosis, cytosolic DNA accumulation, upregulation of IFNβ, IL-6, and NKG2D ligands, lung injury, and pneumonia to a greater extent in older mice than in younger mice (Fig. 8d, e, and Supplementary Fig. 8a–c). Furthermore, Dicer, XPO5, SRSF3, and hnRNPA3 knockdown exaggerated the N protein-induced DNA damage, apoptosis, cytosolic DNA accumulation, upregulation of IFNβ, IL-6, and NKG2D ligands, and pneumonia (Fig. 8f, g and Supplementary Fig. 8d–f). These results suggest that age-related downregulation of Dicer, XPO5, SRSF3, and hnRNPA3 is associated with increased severity of N protein-induced pneumonia.

## PJ34 alleviates N protein-induced pneumonia by preventing Dicer, XPO5, SRSF3, and hnRNPA3 downregulation

PJ34, a poly(ADP-ribose) polymerase (PARP) inhibitor, binds to N protein encoded by different coronaviruses and inhibits its RNA-binding activity[43–45]. PJ34 treatment inhibited the association between SARS-CoV-2 N protein and pre-miRNA and disrupted the interaction between N protein and Dicer, XPO5, SRSF3, and hnRNPA3 (Fig. 9a, b). Although PJ34 did not affect Dicer, XPO5, SRSF3, or hnRNPA3 expression in cells that did not express N protein, it increased their expression in N protein- expressing or SARS-CoV-2-infected cells (Fig. 9c, d, and Supplementary Fig. 9a).

Although PJ34 induced minimal DNA damage in cells that did not express N protein, it alleviated N protein-induced DNA damage (Fig. 9c, d, and Supplementary Fig. 9a, b). PJ34 did not affect protein synthesis in cells not expressing N protein; however, it repressed N protein-induced increase in protein synthesis and relieved N protein-induced proteotoxic stress (Supplementary Fig. 9c, d). Treatment with PJ34 increased the expression of Dicer, XPO5, SRSF3, and hnRNPA3 in SARS-CoV-2-infected lung tissues and ameliorated the pneumonia induced by ectopic N protein expression and SARS-CoV-2 infection (Fig. 9e–g). Although the expression levels of Dicer, XPO5, SRSF3, and

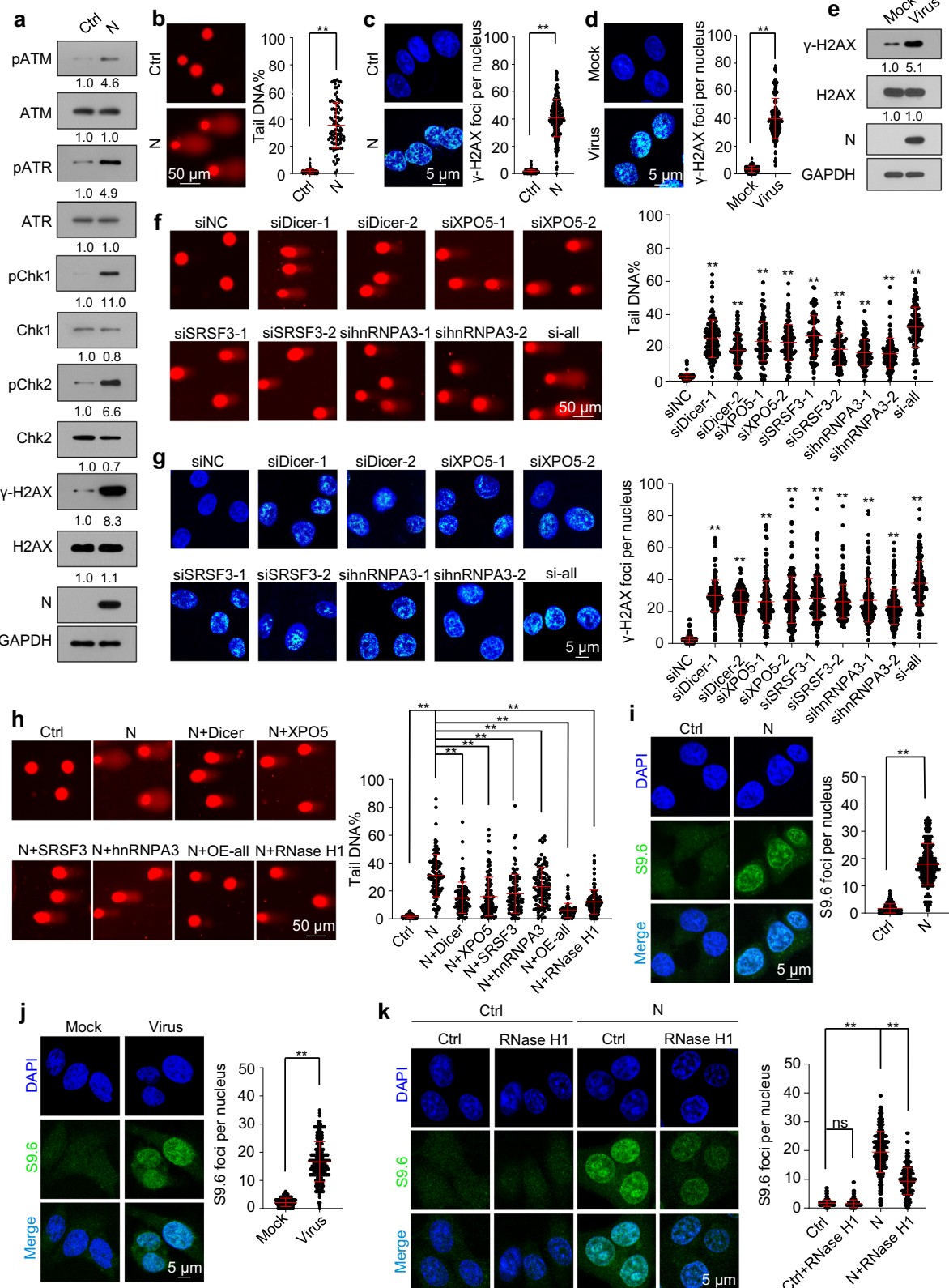

hnRNPA3 were lower in the tissues of old mice than in those of young mice, PJ34 rescued the N-induced downregulation of Dicer, XPO5, SRSF3, and hnRNPA3 and ameliorated N protein-induced pneumonia not only in young mice but also in old mice (Fig. 9g and Supplementary Fig. 9e). Collectively, these results indicate that PJ34 alleviates pneumonia by preventing the N protein-induced downregulation of Dicer, XPO5, SRSF3, and hnRNPA3.

**Anastrozole alleviates N protein-induced pneumonia by promoting Dicer, XPO5, SRSF3, and hnRNPA3 expression**

Treatment with anastrozole, an aromatase inhibitor that enhances Dicer expression[53], increased Dicer, XPO5, SRSF3, and hnRNPA3 expression not only in N protein-expressing or SARS-CoV-2-infected cells, but also in cells not expressing N protein or mock-infected (Fig. 10a, b and Supplementary Fig. 10a). Consistently, anastrozole

**Fig. 3 | SARS-CoV-2 N protein induces DNA damage by downregulating Dicer, XPO5, SRSF3, and hnRNPA3 expression.** See also Supplementary Figs. 3, 4. **a–c** DNA damage in A549-Ctrl or A549-N cells was determined using immunoblotting analysis of the phosphorylation levels of ATM, ATR, Chk1, Chk2, and H2AX (**a**), comet assay (**b**), and immunofluorescence with anti-γ-H2AX antibody (**c**). **d, e** A549-hACE2 cells infected with SARS-CoV-2 or mock were subjected to immunofluorescence with anti-γ-H2AX antibody (**d**) and immunoblotting with indicated antibodies (**e**). A549 cells were transfected with the indicated siRNAs and subjected to comet assay (**f**) and immunofluorescence with anti-γ-H2AX antibody (**g**). **h** A549-Ctrl and A549-N cells transfected with a control plasmid or plasmids overexpressing Dicer, XPO5, SRSF3, hnRNPA3, or RNase H1 were subjected to comet assay. **i** Detection of R-loop in A549-Ctrl or A549-N cells using immunofluorescence with the S9.6 anti-DNA-RNA hybrid antibody. **j** Detection of R-loop in A549-hACE2 cells infected with SARS-CoV-2 or mock using immunofluorescence with the S9.6 anti-DNA-RNA hybrid antibody. **k** Detection of R-loop in A549-Ctrl or A549-N cells transfected with a plasmid expressing RNase H1 or control plasmid using immunofluorescence with the S9.6 anti-DNA-RNA hybrid antibody. Data in (**b–k**) are expressed as mean ± SD (two-tailed unpaired Student's $t$-test). n = 100 (**b, f, h**) or 200 (**c, d, g, i–k**) cells from three biological replicates. **$p < 0.01$; ns not significant ($p > 0.05$; two-tailed unpaired Student's $t$-test). The numbers below the blots (**a, e**) represent the relative densitometric quantification of the bands normalized to corresponding GAPDH bands; the mean values of three independent experiments are shown. Ctrl control plasmid, N N protein, siNC negative control siRNA, si-all cells transfected with siDicer, siXPO5, siSRSF3, and sihnRNPA3 together, OE-all cells transfected with Dicer-, XPO5-, SRSF3-, and hnRNPA3-expressing plasmids together. Source data and exact $p$ values are provided in the Source Data file.

treatment repressed the N protein-induced DNA damage accumulation, protein synthesis, and proteotoxic stress (Fig. 10a, b, and Supplementary Fig. 10a–d) and dose-dependently ameliorated N protein-induced pneumonia (Fig. 10c). The effects of anastrozole on the expression of Dicer, XPO5, SRSF3, and hnRNPA3 and pneumonia were also observed in the SARS-CoV-2-infected mouse model (Fig. 10d, e). Moreover, we found that anastrozole promotes Dicer, XPO5, SRSF3, and hnRNPA3 expression and ameliorates N protein-induced pneumonia not only in young mice but also in old mice (Fig. 10f and Supplementary Fig. 10e). Collectively, these results reveal that anastrozole alleviates SARS-CoV-2 N protein-induced pneumonia by promoting Dicer, XPO5, SRSF3, and hnRNPA3 expression.

## Discussion
DNA damage and proteotoxic stress cause tissue injury by inducing cell death, inflammation, and tissue-destructive immune cell infiltration[7–9,14–16,18,19]. Herein, we demonstrate that SARS-CoV-2 causes pneumonia by inducing DNA damage and proteotoxic stress. Mechanistically, SARS-CoV-2 N protein interacts with Dicer, XPO5, SRSF3, and hnRNPA3, leading to their autophagic degradation. Ectopic N protein expression induces DNA damage by downregulating Dicer, XPO5, SRSF3, and hnRNPA3 expression. As Dicer is essential for 53BP1 recruitment at DSBs[26–29], our findings may explain why N protein impairs 53BP1 recruitment at DSB and hinders nonhomologous end-joining (NHEJ)[54]. Additionally, N protein disturbs proteostasis via two different mechanisms. First, it decreases SRSF3 and hnRPNA3 expression, repressing RNA splicing and inducing the production of insoluble cellular condensates, thereby triggering a proteotoxic stress response[38]. Second, it represses Dicer and XPO5 expression, eliciting a global increase in protein translation owing to global decrease in miRNA levels[25]. Moreover, genetic or pharmacological rescue of Dicer, XPO5, SRSF3, and hnRNPA3 expression relieves the N protein-induced DNA damage, proteotoxic stress, and pneumonia.

Dicer, XPO5, SRSF3, and hnRNPA3 expression was downregulated in the lung tissues of older mice compared with that in younger mice, and N protein induced more severe lung injury and pneumonia in older mice. Dicer, XPO5, SRSF3, and hnRNPA3 knockdown increased the severity of N protein-induced pneumonia. Therefore, age-associated downregulation of Dicer, XPO5, SRSF3, and hnRNPA3 expression in lung tissues may render older individuals more prone to developing severe pneumonia after SARS-CoV-2 infection.

The fundamental function of N protein is to package the viral genomic RNA into a ribonucleocapsid complex and promote viral RNA replication and transcription by recruiting host factors to the viral genome[55]. Our findings indicate that N protein causes pneumonia by inducing DNA damage and proteotoxic stress and may regulate SARS-CoV-2 replication via the following mechanisms. First, N protein suppresses RNAi by decreasing Dicer expression; therefore, it may promote SARS-CoV-2 replication by suppressing the antiviral RNAi[30,31].

Second, as DNA damage response inhibitors block SARS-CoV-2 replication[12], N protein may promote SARS-CoV-2 replication by activating DNA damage response. Third, as splicing inhibition represses SARS-CoV-2 replication[20], N protein may block SARS-CoV-2 replication by inhibiting RNA splicing. Fourth, although N protein inhibits interferon production via interaction with G3BP1[56,57], we observed that it promotes IFNβ expression owing to cytosolic DNA accumulation induced by DNA damage. Therefore, N protein may promote or repress SARS-CoV-2 replication by downregulating or upregulating interferon expression, respectively. Finally, we observed that N protein can induce autophagy, which reportedly promotes or inhibits SARS-CoV-2 replication[58]. Therefore, N protein may regulate SARS-CoV-2 replication in an autophagy-dependent manner.

PJ34, a potent PARP inhibitor, elicits cytotoxicity against human cancer cells without impacting healthy cells[59]. It can also alleviate tissue damage and inflammation with different etiologies[60–62]. PJ34 binds to the coronavirus N protein, inhibiting its RNA-binding activity and hindering viral replication[43–45]. We found that PJ34 disrupted the interaction between the SARS-CoV-2 N protein and Dicer, XPO5, SRSF3, and hnRNPA3 and relieved N protein-induced downregulation of these proteins, thereby alleviating the N protein-induced DNA damage and proteotoxic stress and ultimately mitigating N protein-induced pneumonia. Given that PJ34 can slightly induce DNA damage in cells not expressing N protein, further studies are required to investigate the side effects of PJ34 and to evaluate whether it can be used to treat SARS-CoV-2-induced pneumonia.

Anastrozole, an aromatase inhibitor, is used to treat breast cancer and infertility[63–65]. We recently reported that anastrozole can treat colitis by promoting Dicer expression[53]. Here, we found that anastrozole alleviated N protein- or SARS-CoV-2-induced pneumonia by promoting Dicer, XPO5, SRSF3, and hnRNPA3 expression. As anastrozole can inhibit the conversion of testosterone into estradiol[66], and testosterone increases, while estradiol represses TMPRSS2 expression[67,68], further studies are needed to investigate whether anastrozole can upregulate TMRPSS2 expression and promote SARS-CoV-2 infection. Clinical data have confirmed the safety of long-term administration of low-dose (1 mg per day) anastrozole[63–65]. Moreover, preclinical findings have revealed that short-term administration of high-dose (20 mg/kg) anastrozole is safe[53]. We found that anastrozole dose-dependently ameliorated N protein-induced pneumonia. Therefore, further studies are needed to determine the optimal dosage of anastrozole for COVID-19 treatment.

Although the mRNA levels of *Dicer*, *XPO5*, *SRSF3*, and *hnRNPA3* were associated with COVID-19 severity, and their knockdown increased the severity of N protein-induced pneumonia in mice, whether their expression levels in pre-infected lung tissue are key determinants of COVID-19 severity remains to be elucidated. Therefore, prospective studies are needed to collect human lung tissue samples before SARS-CoV-2 infection and determine whether pre-infection levels of Dicer, XPO5, SRSF3, and hnRNPA3 are associated with COVID-19 severity.

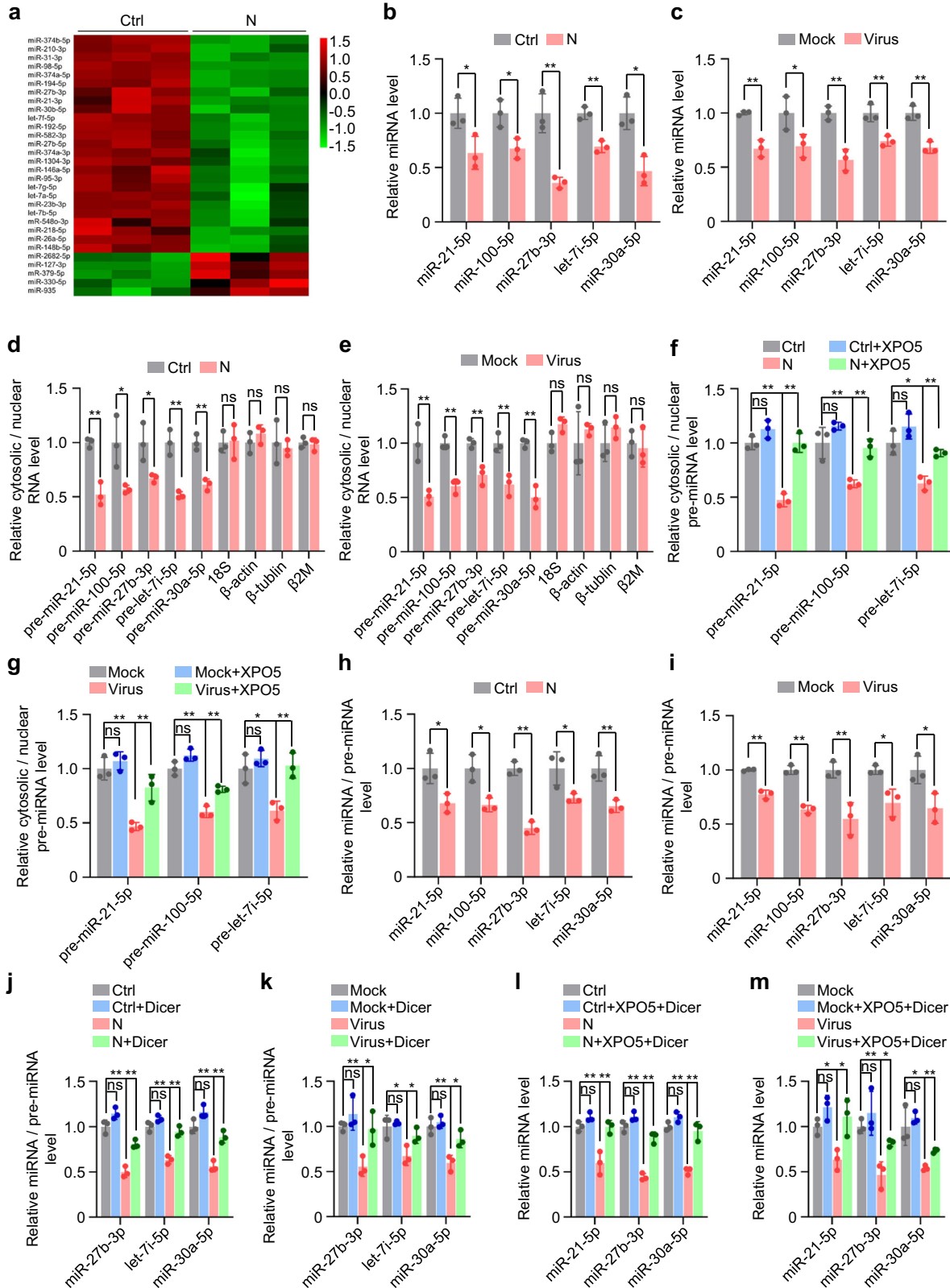

In summary, N protein leads to autophagic degradation of Dicer, XPO5, SRSF3, and hnRNPA3, inducing DNA damage and proteotoxic stress and eventually causing pneumonia. Treatment with PJ34 or anastrozole alleviates the N protein-induced DNA damage, proteotoxic stress, and pneumonia by rescuing Dicer, XPO5, SRSF3, and hnRNPA3 expression. The age-associated downregulation of Dicer, XPO5, SRSF3,

and hnRNPA3 expression in lung tissues is one of the reasons why older individuals are more prone to developing severe pneumonia after SARS-CoV-2 infection than younger ones. Our findings can aid in developing improved treatments for SARS-CoV-2-associated pneumonia and imply that further studies should be conducted to investigate the toxicity and adverse events associated with N protein-based vaccines.

**Fig. 4 | SARS-CoV-2 N protein represses miRNA biogenesis.** See also Supplementary Fig. 5. **a** Heatmap of miRNA expression in A549-Ctrl and A549-N cells based on small RNA sequencing. **b, c** Quantification of five miRNAs in A549-Ctrl or A549-N cells (**b**) or A549-hACE2 cells infected with SARS-CoV-2 or mock (**c**). **d** Ratio of cytosolic levels of different RNAs to their nuclear levels in A549-Ctrl or A549-N cells. **e** Ratio of cytosolic levels of different RNAs to their nuclear levels in A549-hACE2 cells infected with SARS-CoV-2 or mock. **f** Ratio of cytosolic pre-miRNA levels to nuclear pre-miRNA levels in A549-Ctrl or A549-N cells transfected with a plasmid expressing XPO5 or control plasmid. **g** Ratio of cytosolic pre-miRNA levels to nuclear pre-miRNA levels in A549-hACE2 cells transfected with a plasmid expressing XPO5 or control plasmid and infected with SARS-CoV-2 or mock. **h** Ratio of mature miRNA levels to pre-miRNA levels in A549-Ctrl and A549-N cells. **i** Ratio of mature miRNA levels to pre-miRNA levels in A549-hACE2 cells infected with SARS-

CoV-2 or mock. **j** Ratio of mature miRNA levels to pre-miRNA levels in A549-Ctrl and A549-N cells transfected with a plasmid expressing Dicer or a control plasmid. **k** Ratio of mature miRNA levels to pre-miRNA levels in A549-hACE2 cells transfected with a plasmid expressing Dicer or a control plasmid and infected with SARS-CoV-2 or mock. **l** miRNA levels in A549-Ctrl or A549-N cells co-transfected with XPO5-expressing and Dicer-expressing plasmids. **m** miRNA levels in A549-hACE2 cells transfected with XPO5-expressing and Dicer-expressing plasmids or control plasmid and infected with SARS-CoV-2 or mock. Data in (**b–m**) are expressed as mean ± SD of three biological replicates. $**p < 0.01$; $*p < 0.05$; ns not significant ($p > 0.05$; two-tailed unpaired Student's $t$-test). Ctrl control plasmid, N N protein, siNC negative control siRNA. Source data and exact $p$ values are provided in the Source Data file.

## Methods

### Cell lines

Human lung adenocarcinoma epithelial cell line A549 (male, CCL-185), human bronchial epithelial cell line BEAS-2B (male, CRL-9609), human embryonic kidney cell line HEK293T (female, CRL-3216), and African green monkey kidney cell line Vero E6 (female, CRL-1586) were obtained from the American Type Culture Collection (Manassas, VA, USA). A549 and BEAS-2B cells were cultured in Roswell Park Memorial Institute (RPMI)-1640 medium (Hyclone, Logan, UT, USA) supplemented with 10% (v/v) fetal bovine serum (FBS; Hyclone). HEK293T and Vero E6 cells were cultured in Dulbecco's modified Eagle's medium (DMEM; Hyclone) supplemented with 10% (v/v) FBS. All cell lines were cultured at 37 °C in a humidified atmosphere of 5% $CO_2$. All cell lines were routinely tested for *Mycoplasma* contamination using the *Mycoplasma* Detection Kit (Southern Biotech, Birmingham, AL, USA) and authenticated based on polymorphic short-tandem repeat loci. For the reporter gene experiments, HEK293T cells were used owing to their high transfection efficiency; for the reproduction of SARS-CoV-2, Vero E6 cells were used; for other experiments, A549 and BEAS-2B cells were used.

### Viruses

Here, the SARS-CoV-2 Wuhan prototype strain originally isolated from COVID-19 patients was used[29,69]. Cell studies on live SARS-CoV-2 were performed under biosafety level-3 (BSL3) conditions at the Wuhan Institute of Virology, Chinese Academy of Science, or at the Institute of Laboratory Animal Science, Chinese Academy of Medical Sciences & Peking Union Medical College. Mouse studies on live SARS-CoV-2 were performed in an animal biosafety level 3 (ABSL-3) facility with high-efficiency particulate air (HEPA)-filtered isolators at the Institute of Laboratory Animal Science, Chinese Academy of Medical Sciences & Peking Union Medical College.

### Animals

Unless otherwise specified, 8-week-old or 18-month-old C57BL/6J male mice were used in this study. Sex was considered in the analysis as shown in Fig. 7g, showing that N protein induced similar levels of lung injury in male and female mice. Therefore, only male mice were used for all other experiments. Regarding SARS-CoV-2 infection, K18-hACE2 transgenic C57BL/6J (H11-K18-hACE2) mice (GemPharmaTech, Jiangsu, China) were used. Five mice were included per group in each experiment. All mice were housed under pathogen-free conditions in a 12:12-h light/dark cycle, with temperature between 20 and 26 °C and relative humidity between 40 and 70%. The experiments using wild-type mice were performed according to protocols approved by the Institutional Animal Care and Use Committee of Chongqing Medical University (approval number IACUC-CQMU-2024-04052) and Wenzhou Medical University (approval number wydw2022-0593). The experiments involving K18-hACE2 transgenic mice and SARS-CoV-2 infection were conducted in accordance with the protocols approved by the Institutional Animal Care and Use Committee of the Institute of Laboratory

Animal Science, Peking Union Medical College (approval number IACUC-PXZ24001).

### Cell treatment

To block autophagy, cells were treated with chloroquine (20 μM; MedChemExpress, Monmouth Junction, NJ, USA) or bafilomycin A1 (0.2 μM; MedChemExpress) for 12 h. To inhibit proteasome activity, cells were treated with the proteasome inhibitor MG132 (20 μM; MedChemExpress) for 12 h. To investigate the effect of PJ34 on the interaction between N protein and Dicer, XPO5, SRSF3, or hnRNPA3, as well as their expression, cells were treated with PJ34 (MedChemExpress) at indicated concentrations for 2 h. To investigate the effect of anastrozole on the expression of Dicer, XPO5, SRSF3, and hnRNPA3, cells were treated with anastrozole (Aladdin, Shanghai, China) at the indicated concentrations for 24 h. Dimethyl sulfoxide (DMSO) was used as the solvent for chloroquine, MG132, PJ34, and anastrozole and served as the vehicle control. The commercial reagents used in this study are listed in Supplementary Table 1.

### Viral infection

SARS-CoV-2 (multiplicity of infection [MOI] = 0.3) was used in this study. A549 cells stably expressing hACE2 were infected with SARS-CoV-2 for 48 h unless otherwise specified. HEK293T cells stably expressing hACE2 were infected with SARS-CoV-2 for 24 h.

### Mouse treatments

To infect mice with SARS-CoV-2, K18-hACE2 mice were instilled with 50 μL ($2 \times 10^4$ PFU) SARS-CoV-2 or DMEM (mock group). To ectopically express the genes of interest in mouse lung tissues, C57BL/6J mice were intranasally instilled with plasmids (20 μg per mouse) mixed with an in vivo transfection reagent (Entranster™-in vivo, Engreen, Beijing, China) in a total volume of 40 μL. To knock down the genes of interest in mouse lung tissues, mice were intranasally instilled with shRNA plasmids (40 μg per mouse) mixed with the in vivo transfection reagent (Entranster™-in vivo) in a total volume of 40 μL. The plasmids are listed in Supplementary Table 2. To investigate the role of PJ34 or anastrozole in N protein-induced pneumonia, 8-week-old or 18-month-old male C57BL/6J mice were intraperitoneally injected with PJ34 (10 mg/kg) or anastrozole (20 mg/kg or as indicated) 12, 24, and 48 h after plasmid instillation or 24 and 48 h after SARS-CoV-2 infection. The mice were anesthetized using isoflurane (RWD Life Science, Shenzhen, China) and euthanized by cervical dislocation 72 h after plasmid or SARS-CoV-2 instillation, and lung tissues were subjected to further analysis.

To investigate the effects of N and NSP8 viral protein on mouse lung tissues, 8-week-old male mice were randomly allocated to two groups (n = 5/group): group 1 was instilled with a control plasmid and group 2 was instilled with an N protein-expressing plasmid or NSP8 protein-expressing plasmid.

To investigate whether N protein exhibited different effects in male and female mice, 8-week-old male and female mice were allocated to four groups (n = 5/group): group 1, male mice were instilled with

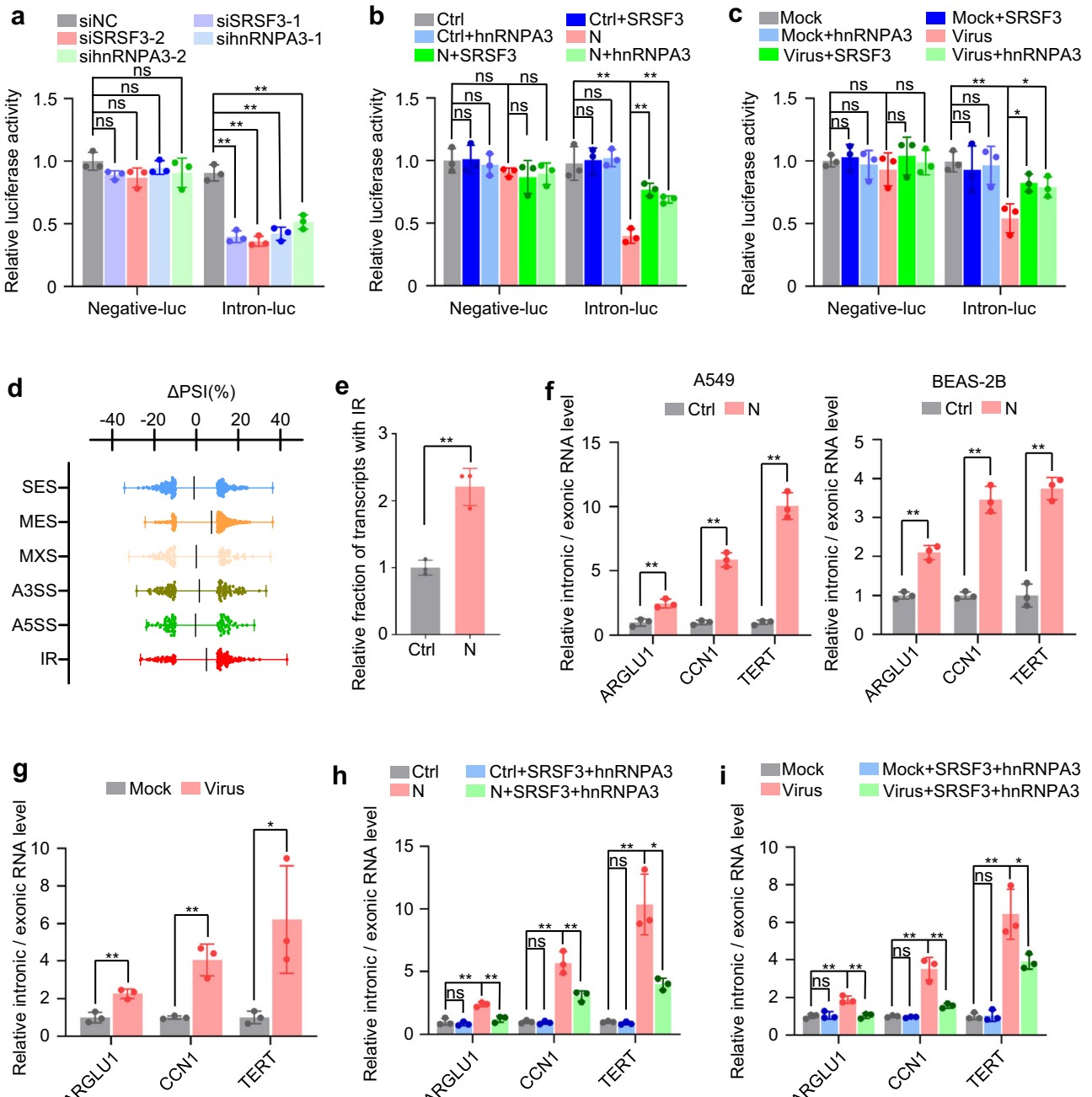

**Fig. 5 | SARS-CoV-2 N protein represses RNA splicing. a** Luciferase assay of HEK293T cells following co-transfection of a splicing reporter with SRSF3-specific siRNA (siSRSF3), hnRNPA3-specific siRNA (sihnRNPA3), or control siRNA. **b** Luciferase assay of HEK293T-Ctrl or HEK293T-N cells following co-transfection of a splicing reporter with SRSF3-expressing, hnRNPA3-expressing, or control plasmids. **c** Luciferase assay of HEK293T-hACE2 cells co-transfected with a splicing reporter with SRSF3-expressing, hnRNPA3-expressing, or control plasmids and then infected with SARS-CoV-2 or mock. **d** Changes in percent spliced-in (PSI) values between A549-Ctrl and A549-N cells were determined using full-length transcriptome sequencing data (n = 3 biological replicates). ΔPSI = PSI(A549-N) − PSI(A549-Ctrl). Each dot represents the ΔPSI of a single gene (error bars, mean ± SD). **e** Relative fraction of transcripts with intron retention over total sequencing reads determined using full-length transcriptome sequencing data. **f** Relative intronic and exonic RNA levels of the indicated genes in A549-Ctrl and A549-N cells or BEAS-2B-Ctrl and BEAS-2B-N cells. **g** Relative intronic and exonic RNA levels of the indicated genes in A549-hACE2 cells infected with SARS-CoV-2 or mock. **h** Relative intronic and exonic RNA levels of the indicated genes in A549-Ctrl and A549-N cells co-transfected with SRSF3-expressing and hnRNPA3-expressing plasmids. **i** Relative intronic and exonic RNA levels of the indicated genes in A549-hACE2 cells co-transfected with SRSF3-expressing and hnRNPA3-expressing plasmids and infected with SARS-CoV-2 or mock. Data in (**a**–**c**, **e**–**i**) are expressed as mean ± SD of three biological replicates. **p < 0.01; *p < 0.05; ns not significant (p > 0.05; two-tailed unpaired Student's t-test). Ctrl control plasmid, N N protein, siNC negative control siRNA, SES single exon skipping, MES multiple exon skipping, MXE mutually exclusive exons, A5SS or A3SS alternative 5' and 3' splice sites, respectively, IR intron retention. Source data and exact p values are provided in the Source Data file.

with control plasmid; group 2, female mice were instilled with control plasmid; group 3, male mice were instilled with N protein-expressing plasmid; group 4, female mice were instilled with N protein-expressing plasmid.

To investigate whether N protein impacted old and young mice differently, 8-week-old and 18-month-old male mice were allocated to four groups (n = 5/group): group 1, 8-week-old mice were instilled with control plasmid; group 2, 18-month-old mice were instilled with

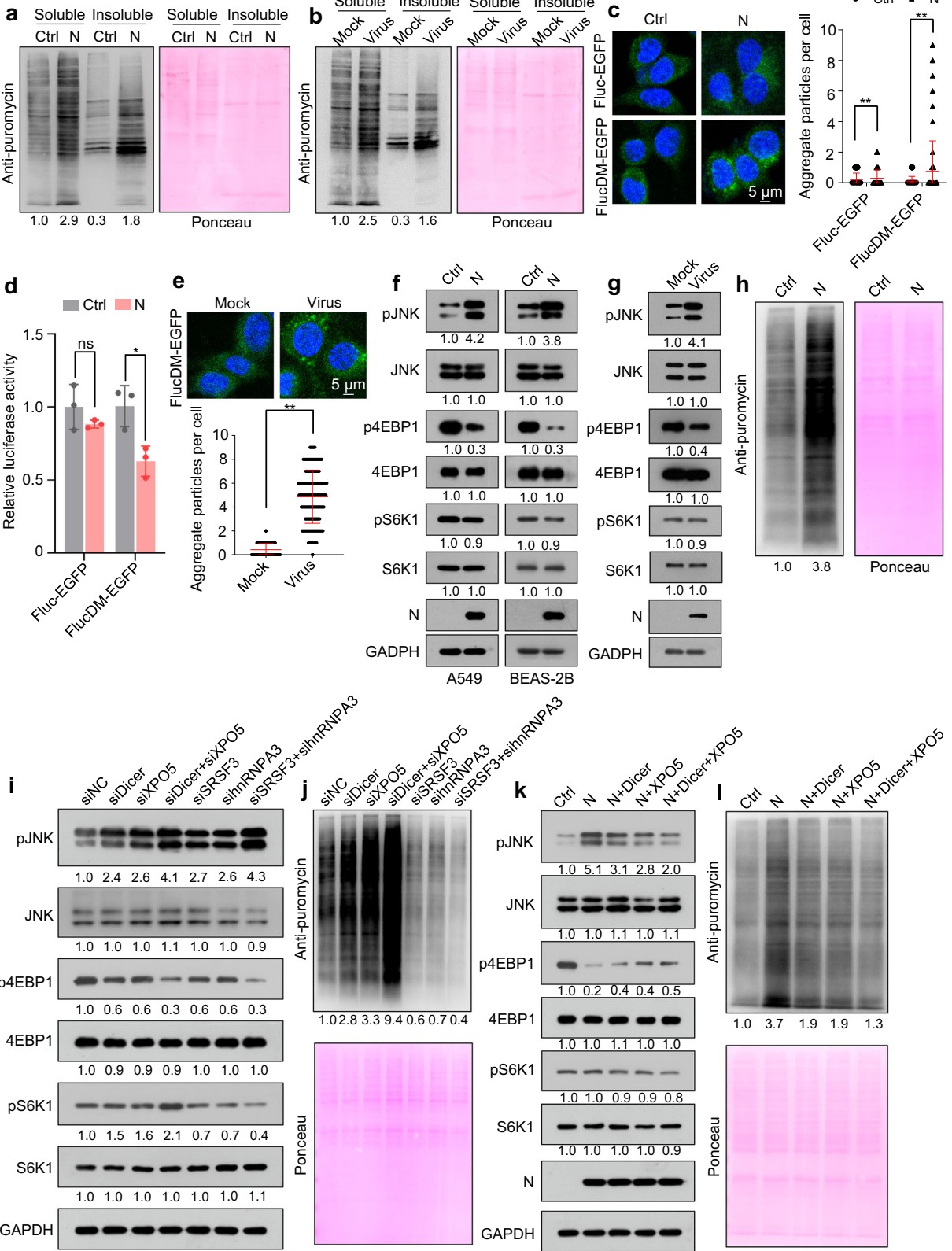

control plasmid; group 3, 8-week-old mice were instilled with N protein-expressing plasmid; and group 4, 18-month-old mice were instilled with N protein-expressing plasmid.

To investigate whether knockdown of Dicer, XPO5, SRSF3, or hnRNPA3 induced lung injury, 8-week-old male mice were allocated to six groups (n = 5/group) and instilled with control shRNA plasmid (shNC), Dicer shRNA plasmid (shDicer), XPO5 shRNA plasmid

(shXPO5), SRSF3 shRNA plasmid (shSRSF3), hnRNPA3 shRNA plasmid (shhnRNPA3), or instilled with all four shRNA plasmids.

To investigate whether knockdown of Dicer, XPO5, SRSF3, or hnRNPA3 increased the severity of N protein-induced pneumonia, 8-week-old male mice were allocated to four groups (n = 5/group): group 1, instilled with control plasmid and shNC; group 2, instilled with N protein-expressing plasmid and shNC; group 3, instilled with control

**Fig. 6 | SARS-CoV-2 N protein induces proteotoxic stress and increases protein translation.** See also Supplementary Fig. 6. Immunoblotting of nascent polypeptides labeled with puromycin in the soluble and insoluble extracts of A549-Ctrl and A549-N cells (**a**) and SARS-CoV-2- or mock-infected A549-hACE2 cells (**b**). A549-Ctrl or A549-N cells transfected with proteotoxic stress sensor reporters encoding Fluc-EGFP or FlucDM-EGFP and analyzed using confocal microscopy (**c**, n = 200 cells from three biological replicates) or a luciferase assay (**d**, n = 3 biological replicates). **e** A549-hACE2 cells transfected with proteotoxic stress sensor reporter FlucDM-EGFP were infected with SARS-CoV-2 and analyzed using confocal microscopy (n = 200 cells from three biological replicates). **f** Immunoblotting of the indicated proteins in A549-Ctrl and A549-N (left) or BEAS-2B-Ctrl and BEAS-2B-N (right) cells. **g** Immunoblotting of the indicated proteins in A549-hACE2 cells infected with SARS-CoV-2 or mock. **h** Immunoblotting of nascent polypeptides labeled with

puromycin in A549-Ctrl and A549-N cells. **i**, **j** Immunoblotting of the indicated proteins (**i**) or nascent polypeptides labeled with puromycin (**j**) in A549 cells transfected with different siRNAs. **k**, **l** Immunoblotting of the indicated proteins (**k**) or nascent polypeptides labeled with puromycin (**l**) in A549-Ctrl, A549-N, and A549-N cells transfected with Dicer-expressing plasmid, XPO5-expressing plasmid, or both. Ponceau S staining image represents the loading control in (**a**, **b**, **h**, **j**, **l**). The numbers below the blots indicate the relative densitometric quantification of the bands normalized to corresponding GAPDH bands (**f**, **g**, **i**, **k**) or Ponceau S staining bands (**a**, **b**, **h**, **j**, **l**); the mean values of three independent experiments are shown. Data in (**c**–**e**) are expressed as mean ± SD. \*\*$p < 0.01$; \*$p < 0.05$; ns not significant ($p > 0.05$; two-tailed unpaired Student's $t$-test). Ctrl control plasmid, N N protein, siNC negative control siRNA. Source data and exact $p$ values are provided in the Source Data file.

---

plasmid, shDicer, shXPO5, shSRSF3, and shhnRNPA3; and group 4, instilled with N protein-expressing plasmid, shDicer, shXPO5, shSRSF3, and shhnRNPA3.

To investigate whether the overexpression of Dicer, XPO5, SRSF3, and hnRNPA3 alleviated N protein-induced pneumonia, 8-week-old male mice were allocated to four groups (n = 5/group): group 1, instilled with control plasmid; group 2, instilled with N protein-expressing plasmid; group 3, instilled with control plasmid, Dicer-, XPO5-, SRSF3-, and hnRNPA3-expressing plasmids; and group 4, instilled with N protein-, Dicer-, XPO5-, SRSF3-, and hnRNPA3-expressing plasmids.

To investigate whether the overexpression of Dicer, XPO5, SRSF3, and hnRNPA3 alleviated SARS-CoV-2-induced pneumonia, 8-week-old K18-hACE2 mice were allocated to four groups (n = 5/group): group 1, instilled with control plasmid and DMEM; group 2, instilled with control plasmid and SARS-CoV-2; group 3, instilled with Dicer-, XPO5-, SRSF3-, and hnRNPA3-expressing plasmids and DMEM; and group 4, instilled with N protein-, Dicer-, XPO5-, SRSF3-, and hnRNPA3-expressing plasmids and SARS-CoV-2.

To investigate whether PJ34 alleviated N protein-induced pneumonia, 8-week-old or 18-month-old male mice were allocated to four groups (n = 5/group): group 1, instilled with control plasmid and intraperitoneally injected with DMSO; group 2, instilled with control plasmid and intraperitoneally injected with PJ34; group 3, instilled with N protein-expressing plasmid and intraperitoneally injected with DMSO; and group 4, instilled with N protein-expressing plasmid and intraperitoneally injected with PJ34.

To investigate whether PJ34 alleviated SARS-CoV-2-induced pneumonia, 8-week-old K18-hACE2 mice were allocated to four groups (n = 5/group): group 1, instilled with DMEM and intraperitoneally injected with DMSO; group 2, instilled with DMEM and intraperitoneally injected with PJ34; group 3, instilled with SARS-CoV-2 and intraperitoneally injected with DMSO; and group 4, instilled with SARS-CoV-2 and intraperitoneally injected with PJ34.

To investigate whether anastrozole alleviated N protein-induced pneumonia, 8-week-old or 18-month-old male mice were allocated to four groups (n = 5/group): group 1, instilled with control plasmid and intraperitoneally injected with DMSO; group 2, instilled with control plasmid and intraperitoneally injected with anastrozole; group 3, instilled with N protein-expressing plasmid and intraperitoneally injected with DMSO; and group 4, instilled with N protein-expressing plasmid and intraperitoneally injected with anastrozole.

To investigate the dosage-dependent effect of anastrozole on N protein-induced pneumonia, 8-week-old male mice were allocated to eight groups (n = 5/group): group 1, instilled with control plasmid and intraperitoneally injected with DMSO; groups 2–4, instilled with control plasmid and intraperitoneally injected with 0.2 mg/kg, 2 mg/kg, and 20 mg/kg anastrozole, respectively; group 5, instilled with N protein-expressing plasmid and intraperitoneally injected with DMSO; and groups 6–8, instilled with N protein-expressing plasmid and

intraperitoneally injected with 0.2 mg/kg, 2 mg/kg, and 20 mg/kg anastrozole, respectively.

To investigate whether anastrozole alleviated SARS-CoV-2-induced pneumonia, 8-week-old K18-hACE2 mice were allocated to four groups (n = 5/group): group 1, instilled with DMEM and intraperitoneally injected with DMSO; group 2, instilled with DMEM and intraperitoneally injected with anastrozole; group 3, instilled with SARS-CoV-2 and intraperitoneally injected with DMSO; and group 4, instilled with SARS-CoV-2 and intraperitoneally injected with anastrozole.

### Histological analysis of acute lung injury

To assay lung injury, the lung tissues were fixed in 4% formaldehyde, dehydrated, embedded in paraffin, and sectioned at 4 μm thickness. Then they were stained with hematoxylin and eosin (HE) according to standard protocols, dehydrated in a graded series of ethanol, cleared in xylene, cover-slipped, and photographed using a light microscope (Axioscope 5; Carl Zeiss, Jena, Germany).

Acute lung injury (ALI) scoring was conducted by two pathologists blinded to the treatment protocol using the following parameters as previously reported[70,71]: (1) septal mononuclear cell/lymphocyte/neutrophil/macrophage infiltration; (2) septal hemorrhage and congestion and alveolar hemorrhage; (3) alveolar edema and proteinaceous debris filling the airspace; (4) alveolar septal thickening. The severity of each category was graded from 0 (minimal) to 3 (maximal), and the total score was calculated by adding the scores for each category.

### Immunohistochemistry and immunofluorescence

Heat-induced antigen retrieval on lung tissue sections was performed in antigen retrieval solution. Endogenous peroxidase activity was blocked using 3% $H_2O_2$ (ZSGB-BIO, Beijing, China) for 10 min, followed by incubation of the sections with the blocking solution (ZSGB-BIO) for 15 min to minimize non-specific staining. The sections were incubated overnight with anti-CD3 (17617-1-AP; Proteintech, Chicago, IL, USA; 1:200), anti-CD22 (21894-1-AP; Proteintech; 1:2000), anti-CD68 (28058-1-AP; Proteintech; 1:1000), anti-Ly6G (ab238132; Abcam, Boston, MA, USA; 1:100), and anti-dsDNA (MAB1293; Sigma-Aldrich, St. Louis, MO, USA; 1:500) at 4 °C.

For immunohistochemistry, the sections were washed three times with phosphate-buffered saline (PBS) to remove unbound primary antibodies, incubated with Elivision™ super HRP (MXB Biotechnologies, Fuzhou, China), and visualized using a DAB kit (MXB Biotechnologies) according to the manufacturer's instructions. The sections were counterstained with hematoxylin, dehydrated in a graded ethanol series, cleared in xylene, cover-slipped, and photographed under a light microscope (Axioscope 5; Carl Zeiss).

For immunofluorescence, the sections were washed three times with PBS to remove unbound primary antibodies and then incubated with goat anti-rabbit-Alexa Fluor™488 (A-11008; Thermo Fisher Scientific, Waltham, MA, USA) at 37 °C for 30 min, counterstained with 200 ng/mL 4′,6-diamidino-2-phenylindole (DAPI; Sigma-Aldrich) for

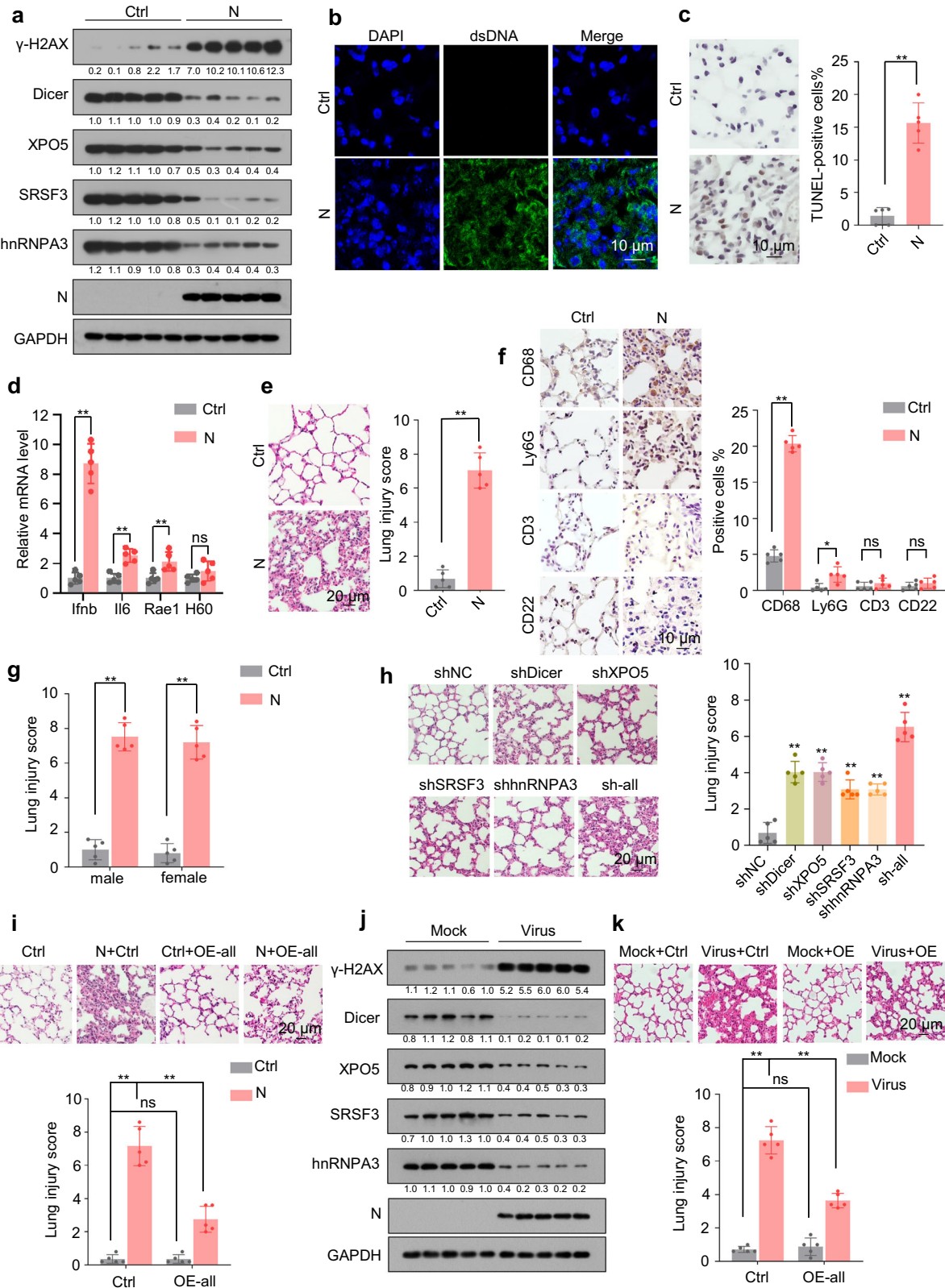

5 min to visualize nuclei, and imaged under a confocal microscope (LEICA DMi8; Leica Microsystems, Wetzlar, Germany).

**Terminal deoxynucleotidyl transferase dUTP nick end labeling (TUNEL) assay**
To analyze apoptosis in mouse lung tissues, hydrated lung sections were evaluated using a TUNEL assay kit (Beyotime

Biotechnology, Shanghai, China) according to the manufacturer's instructions and photographed under a light microscope (Axioscope 5; Carl Zeiss).

**DNA and siRNA transfection**
Cells were transfected with plasmids and siRNAs using Lipofectamine 2000 (Thermo Fisher Scientific) according to the manufacturer's

**Fig. 7 | SARS-CoV-2 N protein induces pneumonia in mice.** See also Supplementary Fig. 7. **a**–**f** Eight-week-old male mice were intranasally instilled with control- or N protein-expressing plasmid, and lung tissues were subjected to immunoblotting with the indicated antibodies (**a**, n = 5 mice per group), immunofluorescence with anti-dsDNA antibody (**b**); terminal deoxynucleotidyl transferase dUTP nick end labeling (TUNEL) assay (**c**); RT-qPCR analysis of *Ifnb*, *Il6*, *Rae1*, and *H60* expression (**d**); hematoxylin and eosin (HE) staining to assess lung injury (**e**); and immunohistochemistry with antibodies against markers of different immune cells, including CD68 (macrophages), Ly6G (neutrophils), CD3 (T cells), and CD22 (B cells; **f**). Representative images of immunofluorescence with anti-dsDNA antibody in lung tissues are shown (n = 5 mice per group). **g** Eight-week-old male or female mice were intranasally instilled with control- or N protein-expressing plasmid, and their lung tissues were subjected to HE staining to assess lung injury. **h** Mice were intranasally instilled with plasmids expressing different shRNAs, and lung tissues were subjected to HE staining to assess lung injury. **i** Mice were intranasally instilled with N protein-expressing plasmid or co-instilled with N protein-expressing plasmid and plasmids expressing Dicer, XPO5, SRSF3, and hnRNPA3, and their lung tissues were subjected to HE staining to assess lung injury. **j** Eight-week-old K18-hACE2 mice were intranasally instilled with SARS-CoV-2 or mock-instilled, and their lung tissues were subjected to immunoblotting with the indicated antibodies (n = 5 mice per group). **k** Eight-week-old K18-hACE2 mice intranasally instilled with plasmids expressing Dicer, XPO5, SRSF3, and hnRNPA3 were intranasally instilled with SARS-CoV-2 or mock-instilled, and their lung tissues were subjected to HE staining to assess lung injury. The numbers below the blots (**a**, **j**) indicate the relative densitometric quantification of the bands normalized to corresponding GAPDH bands. Data in (**c**–**i**, **k**) are expressed as mean ± SD of five mice. **p < 0.01; *p < 0.05; ns not significant (p > 0.05; two-tailed unpaired Student's t-test. Ctrl control plasmid, N N protein, shNC negative control shRNA, sh-all mice instilled with shDicer, shXPO5, shSRSF3, and shhnRNPA3 together, OE-all mice instilled with Dicer-, XPO5-, SRSF3-, and hnRNPA3-expressing plasmids together. Source data and exact p values are provided in the Source Data file.

---

instructions. The plasmids and siRNAs used are listed in Supplementary Tables 2 and 3.

## Lentivirus preparation and transduction

To prepare lentiviruses, HEK293T cells were transiently transfected with pLVX-EF1alpha-2xStrep-IRES-Puro (control plasmid), pLVX-EF1alpha-SARS-CoV-2-N-2×Strep-IRES-Puro (N protein-expressing plasmid), or pLVX-3×FLAG-hACE2 (hACE2-expressing plasmid), together with pMD2.G (Addgene, Cambridge, MA, USA) and psPAX2 (Addgene) using Lipofectamine 2000 according to the manufacturer's recommendations. Seventy-two hours post-transfection, the supernatant was passed through a 0.45-μm syringe filter unit (EMD Millipore, Burlington, MA, USA). HBLV-EGFP-LC3-PURO lentivirus was purchased from Hanbio Biotechnology (Shanghai, China).

For transduction, A549, BEAS-2B, and HEK293T cells at ~50% confluency were transduced with lentivirus (MOI = 0.3) in the presence of 4 μg/mL polybrene (Sigma-Aldrich). Forty-eight hours post-infection, the cells were cultured in the presence of 2 μg/mL puromycin to select for transduced cells. The cells transduced with the control lentivirus were designated as A549-Control (A549-Ctrl), BEAS-2B-Control (BEAS-2B-Ctrl), and HEK293T-Control (HEK293T-Ctrl) cells. The cells transduced with N protein-expressing lentivirus were designated as A549-N protein (A549-N), BEAS-2B-N protein (BEAS-2B-N), and HEK293T-N protein (HEK293T-N) cells. The cells infected with the hACE2 protein-expressing lentivirus were designated A549-hACE2 and HEK293T-hACE2 cells.

## Comet assay

The comet assay was performed as previously reported[24]. Briefly, cells were washed, trypsinized, and suspended in 37 °C PBS at a concentration of $2 \times 10^5$ cells/mL. A low-melting-point agarose (0.7% [w/v]; Solarbio, Beijing, China) suspension at 37 °C was then added to the cells at a 4:1 volume ratio and immediately transferred to a slide pre-coated with 0.6% (w/v) regular agarose (Solarbio). The slides were coverslipped and incubated for 10 min at 4 °C. Subsequently, the coverslip was gently removed, and the slide containing the cells was immersed in prechilled lysis solution (2.5 M NaCl, 100 mM ethylenediamine tetraacetic acid [EDTA], 10 mM Tris-HCl with 10% [v/v] DMSO, and 1% [v/v] Triton X-100 freshly added, pH 10.0) for 90 min at 4 °C. After lysis, the slides were washed with prechilled electrophoretic solution (1 mM EDTA, 300 mM NaOH, pH > 13.0), incubated in electrophoretic solution for 30 min at 4 °C, and subjected to electrophoresis at 25 V for 30 min at 4 °C in the dark. The slides were neutralized with 400 mM Tris (pH 7.5) for 30 min at 4 °C, dehydrated with 75% (v/v), 95% (v/v), and absolute ethanol for 5 min each at 25 °C, and stained with 0.1 μg/mL ethidium bromide (Sigma-Aldrich) for 30 min in the dark. Comet images were visualized using a fluorescence microscope (LEICA DMI3000 B; Leica Microsystems) and analyzed using CASP v1.2.3b2 (CaspLab, Wroclaw, Poland). The percentage of DNA in tail (Tail DNA%) was measured in 30–50 cells in an individual biological replicate, and the total number of cells in the three biological replicates was 100.

## Cytoplasmic/nuclear RNA extraction

Cytosolic and nuclear RNAs were extracted using a Cytoplasmic & Nuclear RNA Purification Kit (Norgen Biotek, Thorold, ON, USA) according to the manufacturer's protocol.

## Luciferase reporter assay

HEK293T-hACE2 cells were prepared by transfecting the pLVX-3×FLAG-hACE2 plasmid into HEK293T cells. To examine the effect of SARS-CoV-2 on RNA silencing, HEK293T-hACE2 cells were seeded in a 48-well plate ($2 \times 10^4$ cells per well) and co-transfected with 20 ng pRL-CMV (Promega, Madison, WI, USA), 100 ng pGL3-Control vector (Promega), and 100 ng pLKO.1-shNC-puro (shNC) or pLKO.1-shFluc-puro (shFluc). The cells were then infected with SARS-CoV-2 24 h post-transfection, and luciferase assays were performed 24 h post-infection.

To examine the effect of SARS-CoV-2 on RNA splicing, HEK293T-hACE2 cells in a 48-well plate were co-transfected with 20 ng pRL-CMV (Promega) and 100 ng CMV-LUC2CP/ARE (Negative-luc) or CMV-LUC2CP/intron/ARE (Intron-luc). The cells were then infected with SARS-CoV-2 24 h post-transfection, and luciferase assays were performed 24 h post-infection.

To assess the effect of SARS-CoV-2-encoding proteins on RNA silencing, HEK293T cells in a 48-well plate were co-transfected with 20 ng pRL-CMV, 100 ng pGL3-Control vector (Promega), and 100 ng shNC or shFluc, together with pLVX-EF1alpha-2xStrep-IRES-Puro (Ctrl), pLVX-EF1alpha-SARS-CoV-2-S-2xStrep-IRES-Puro (S), pLVX-EF1alpha-SARS-CoV-2-N-2xStrep-IRES-Puro (N), pLVX-EF1alpha-SARS-CoV-2-M-2xStrep-IRES-Puro (M), or pLVX-EF1alpha-SARS-CoV-2-E-2xStrep-IRES-Puro (E); luciferase assays were performed 48 h post-transfection.

To assess the effect of SARS-CoV-2-encoding proteins on RNA splicing, HEK293T cells in a 48-well plate were co-transfected with 20 ng pRL-CMV and 100 ng Negative-luc or Intron-luc, together with plasmids encoding S, N, M, or E; luciferase assays were performed 48 h post-transfection.

To assess the effect of N protein on RNA silencing, HEK293T-Ctrl and HEK293T-N cells in a 48-well plate were co-transfected with 20 ng pRL-CMV (Promega), 100 ng pGL3-Control vector (Promega), and 100 ng shNC or shFluc; luciferase assays were performed 48 h post-transfection.

To examine the effect of N protein on RNA splicing, HEK293T-Ctrl and HEK293T-N cells in a 48-well plate were co-transfected with 20 ng pRL-CMV (Promega) and 100 ng Negative-luc or Intron-luc; luciferase assays were performed 48 h post-transfection.

To examine whether SRSF3 and hnRNPA3 overexpression can rescue the inhibitory effect of N protein on RNA splicing, HEK293T-Ctrl

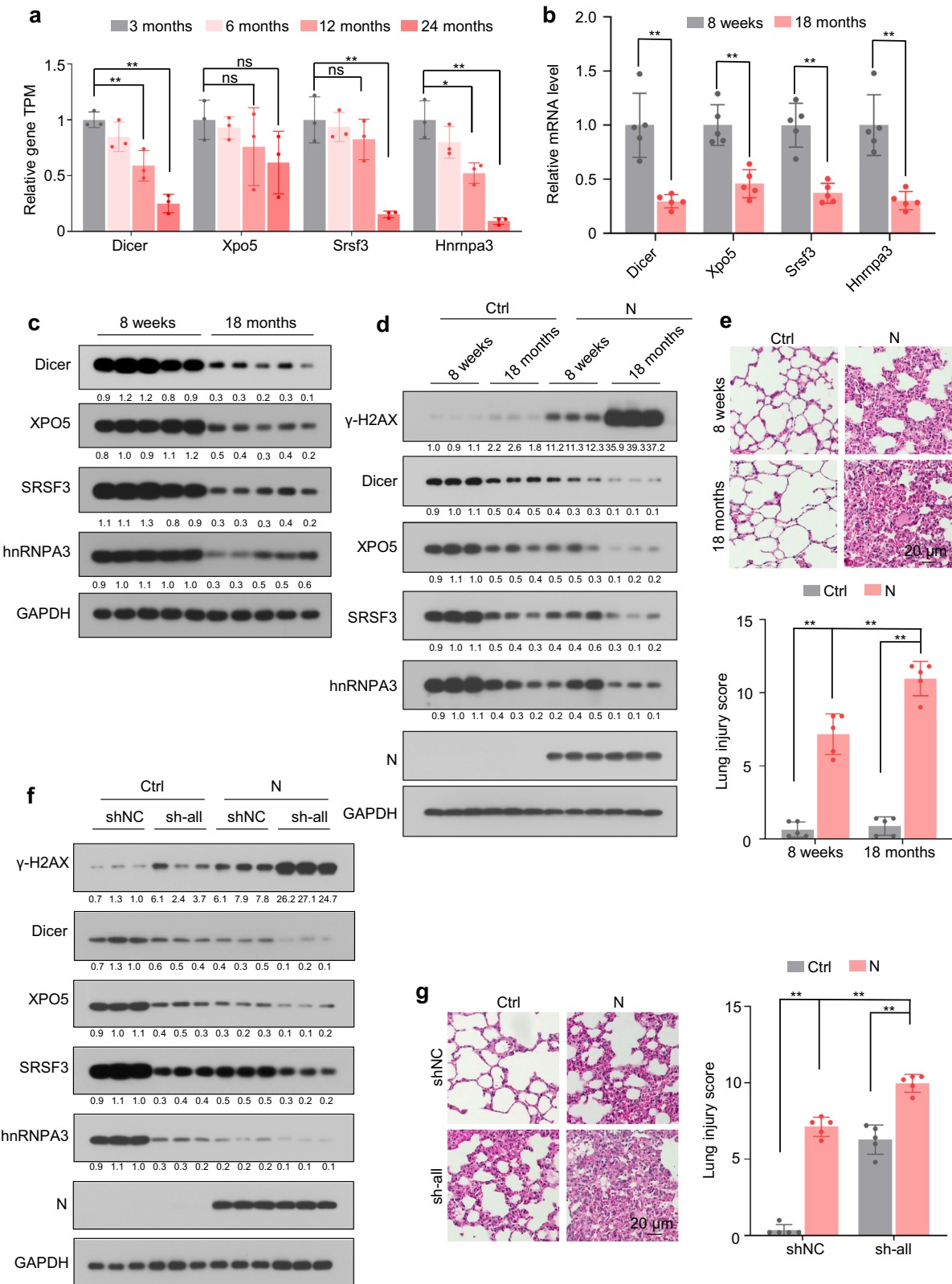

or HEK293T-N cells in a 48-well plate were co-transfected with 20 ng pRL-CMV (Promega) and 100 ng Negative-luc or Intron-luc, together with pCDH-CMV-MCS-EF1-Puro-SRSF3 (SRSF3) or pCDH-CMV-MCS-EF1-Puro-hnRNPA3 (hnRNPA3); luciferase assays were performed 48 h post-transfection.

To examine whether SRSF3 and hnRNPA3 overexpression can rescue the inhibitory effect of SARS-CoV-2 on RNA splicing, HEK293T-

hACE2 cells were co-transfected in a 48-well plate with 20 ng pRL-CMV (Promega) and 100 ng CMV-LUC2CP/ARE (Negative-luc) or CMV-LUC2CP/intron/ARE (Intron-luc). The cells were then infected with SARS-CoV-2 48 h post-transfection, and luciferase assays were performed 24 h post-infection.

To evaluate the effect of N protein on proteotoxic stress, A549-Ctrl or A549-N cells were co-transfected in a 48-well plate with 20 ng

**Fig. 8 | Age-related downregulation of Dicer, XPO5, SRSF3, and hnRNPA3 is associated with the severity of N protein-induced pneumonia.** See also Supplementary Fig. 8. **a** Expression levels of *Dicer*, *Xpo5*, *Srsf3*, and *Hnrnpa3* in the lung tissues of 3-, 6-, 12-, and 24-month-old mice. Data were downloaded from the Gene Expression Omnibus (GEO) database (GSE209891) and are expressed as mean ± SD of three mice per group (two-tailed unpaired Student's *t*-test). **b, c** mRNA (**b**) and protein (**c**, n = 5 mice per group) levels of Dicer, XPO5, SRSF3, and hnRNPA3 in the lung tissues of 8-week-old and 18-month-old mice. **d, e** Mice (8-week-old and 18-month-old) were intranasally instilled with control or N protein-expressing plasmid and subjected to immunoblotting with the indicated antibodies (**d**) and HE staining to assess lung injury (**e**). **f, g** Eight-week-old mice were intranasally instilled with N

protein-expressing plasmid or co-instilled with N protein-expressing plasmid and shRNA-expressing plasmids, and their lung tissues were subjected to immuno-blotting with the indicated antibodies (**f**) and HE staining to assess lung injury (**g**). The numbers below the blots (**c**, **d**, **f**) indicate the relative densitometric quantification of the bands normalized to corresponding GAPDH bands. Representative immunoblots of lung tissues from three out of five mice per group are shown (**d**, **f**). Data in (**b**, **e**, **g**) are expressed as mean ± SD of five mice. \*\*$p < 0.01$; \*$p < 0.05$; ns not significant ($p > 0.05$; two-tailed unpaired Student's *t*-test). Ctrl control plasmid, N N protein, shNC negative control shRNA, sh-all mice instilled with shDicer, shXPO5, shSRSF3, and shhnRNPA3 together. Source data and exact *p* values are provided in the Source Data file.

pRL-CMV (Promega) and 200 ng pCI-neo Fluc-EGFP or pCI-neo FlucDM-EGFP; luciferase assays were performed 48 h post-transfection.

Luciferase activity was measured using a dual-luciferase reporter assay system (Promega) according to the manufacturer's protocol. pRL-CMV expressing *Renilla* luciferase under the control of the *CMV* promoter was used as an external control to correct for differences in transfection and harvest efficiencies.

### Immunoblotting

Cells and tissues were lysed in radioimmunoprecipitation assay buffer supplemented with protease inhibitor cocktail tablets (Roche, Basel, Switzerland) and phosphatase inhibitor cocktail (APExBIO Technology LLC, Houston, TX, USA) at 4 °C. Protein concentration was measured using a Bicinchoninic Acid (BCA) Protein Assay Kit (Beyotime Biotechnology). Equal amounts of proteins were resolved by sodium dodecyl sulfate (SDS) polyacrylamide gel electrophoresis and transferred onto a polyvinylidene fluoride membrane (Bio-Rad Laboratories, Hercules, CA, USA). The protein amount on the membrane was detected using Ponceau S Staining Solution (Thermo Fisher Scientific). Membranes were blocked with 5% bovine serum albumin prepared in TBST (20 mM Tris-HCl [pH 7.5], 150 mM NaCl, and 0.1% Tween 20) for 1 h and subsequently probed with the indicated primary antibodies listed in Supplementary Table 4, followed by incubation with appropriate horseradish peroxidase-conjugated secondary antibodies (Supplementary Table 4). The blots were then treated with Clarity Western Enhanced Chemiluminescence Substrate (Bio-Rad Laboratories) and exposed to X-ray films (FujiFilm, Tokyo, Japan) in a dark room or visualized using the ChampChemi imaging system (Beijing Sage Creation Science, Beijing, China).

### Reverse transcription-quantitative polymerase chain reaction (RT-qPCR)

Total RNA was prepared using TRIzol reagent (Life Technologies, Waltham, MA, USA) and then subjected to gel electrophoresis to assess RNA quality. Purified RNA was reverse-transcribed using the HiScript III RT SuperMix for qPCR (+gDNA wiper) kit (Vazyme Biotech Co., Ltd., Nanjing, China) for mRNAs and the miDETECT A Track miRNA qRT-PCR Starter Kit (RiboBio, Guangzhou, China) for miRNAs and pre-miRNAs. SYBR Green-based real-time PCR was performed using the ChamQ Universal SYBR qPCR Master Mix (Vazyme Biotech Co., Ltd.) and the ABI 7500 FAST sequence detection system (Life Technologies). The expression levels in all samples were normalized to those of *GAPDH* for mRNAs and *U6* for miRNAs and pre-miRNAs unless otherwise specified. For calculating the ratio of nuclear to cytosolic pre-miRNAs or 18S rRNA (Fig. 4d–g and Supplementary Fig. 5c, d), *GAPDH* and *U6* were used as cytosolic and nuclear internal controls, respectively, as previously reported[72,73]. For calculation of the ratio between nuclear and cytosolic *GAPDH*, *β-actin*, *β-tubulin*, *β2M*, or *U6* (Fig. 4d, e), 18S rRNA was used as the cytosolic or nuclear internal control[74,75]. Primer sequences are listed in Supplementary Table 3.

### Fluorescein isothiocyanate (FITC)-Annexin V/propidium iodide apoptosis assay

Apoptosis was detected via flow cytometry using an FITC Annexin V Apoptosis Detection Kit (BD Biosciences, San Jose, CA, USA), according to the manufacturer's instructions. FSC-A/SSC-A gate was set in order to exclude cell debris, and SSC-H/SSC-A and the FSC-H/FSC-A gate was then set in order to exclude doublets. The gating strategy is shown in Supplementary Fig. 7.

### Immunoprecipitation (IP)

Cells were lysed with IP binding buffer (20 mM Tris [pH 7.5], 150 mM NaCl, and 1% [v/v] Triton X-100) supplemented with protease inhibitor cocktail tablets (Roche) at 4 °C for 30 min with continuous rotation, followed by centrifugation at 13,000 × *g* for 10 min. Equal amounts of the lysate were incubated overnight with an anti-nucleocapsid protein antibody (40588-RC02; Sino Biological, Beijing, China), anti-p62 (SQSTM1) antibody (18420-1-AP; Proteintech), or normal rabbit immunoglobulin G (IgG; A7058; Beyotime Biotechnology) at 4 °C and the antibody-protein complexes were sequestered using Protein A/G magnetic beads (Selleck Chemicals, Houston, TX, USA). The beads were washed five times with IP washing buffer (50 mM Tris [pH 7.5], 150 mM NaCl, 1% [v/v] Triton X-100), supplemented with protease inhibitor cocktail tablets (Roche), and eluted with SDS loading buffer at 100 °C for 10 min. The pulled-down immune complexes were subjected to immunoblotting to identify the interacting proteins.

To determine whether RNA is involved in the interaction between N protein and Dicer, XPO5, SRSF3, hnRNPA3, or p62, the cellular extracts were treated with RNase A (1 mg/mL; TakaRa, Dalian, China), RNase T1 (20 U/mL; Thermo Fisher Scientific), RNase V1 (20 U/mL; Life Technologies), and RNase I (20 U/mL; Life Technologies) for 15 min at 37 °C before IP. To determine whether inhibiting the RNA-binding activity of N protein disrupted the interaction between N protein and Dicer, XPO5, SRSF3, or hnRNPA3, cells were treated with PJ34 (50 μM) for 2 h before being lysed.

### RNA-immunoprecipitation (RIP) assay

Cells were cross-linked using 1% (v/v) formaldehyde. Crosslinking was stopped by adding 200 mM glycine, and the cells were lysed in 0.5 mL ice-cold RIP lysis buffer (50 mM Tris-HCl [pH 8.0], 150 mM KCl, 5 mM EDTA, 0.1% [m/v] SDS, 1% [v/v] Triton X-100, 0.5% [m/v] sodium deoxycholate, and 0.5 mM dithiothreitol [DTT]) supplemented with 500 U/mL RNase inhibitor (Vazyme Biotech Co., Ltd) and protease inhibitor cocktail tablets (Roche) and sonicated at 4 °C for 4 min (30 s on and 30 s off) on a QSonica Q800R3 sonicator (QSonica, Newton, CT, USA) at 30% power. The resulting cell lysate was centrifuged at 12,000 × *g* and 4 °C for 10 min. The supernatants were mixed with equal volumes of RIP binding buffer (25 mM Tris-HCl [pH 7.5], 150 mM KCl, 5 mM EDTA, 0.5% [v/v] NP-40, and 0.5 mM DTT) supplemented with 500 U/mL RNase inhibitor (Vazyme Biotech Co., Ltd) and protease inhibitor cocktail tablets (Roche) and incubated with an anti-nucleocapsid protein antibody (Sino Biological) or normal rabbit IgG control (Beyotime Biotechnology) at 4 °C for 6 h.

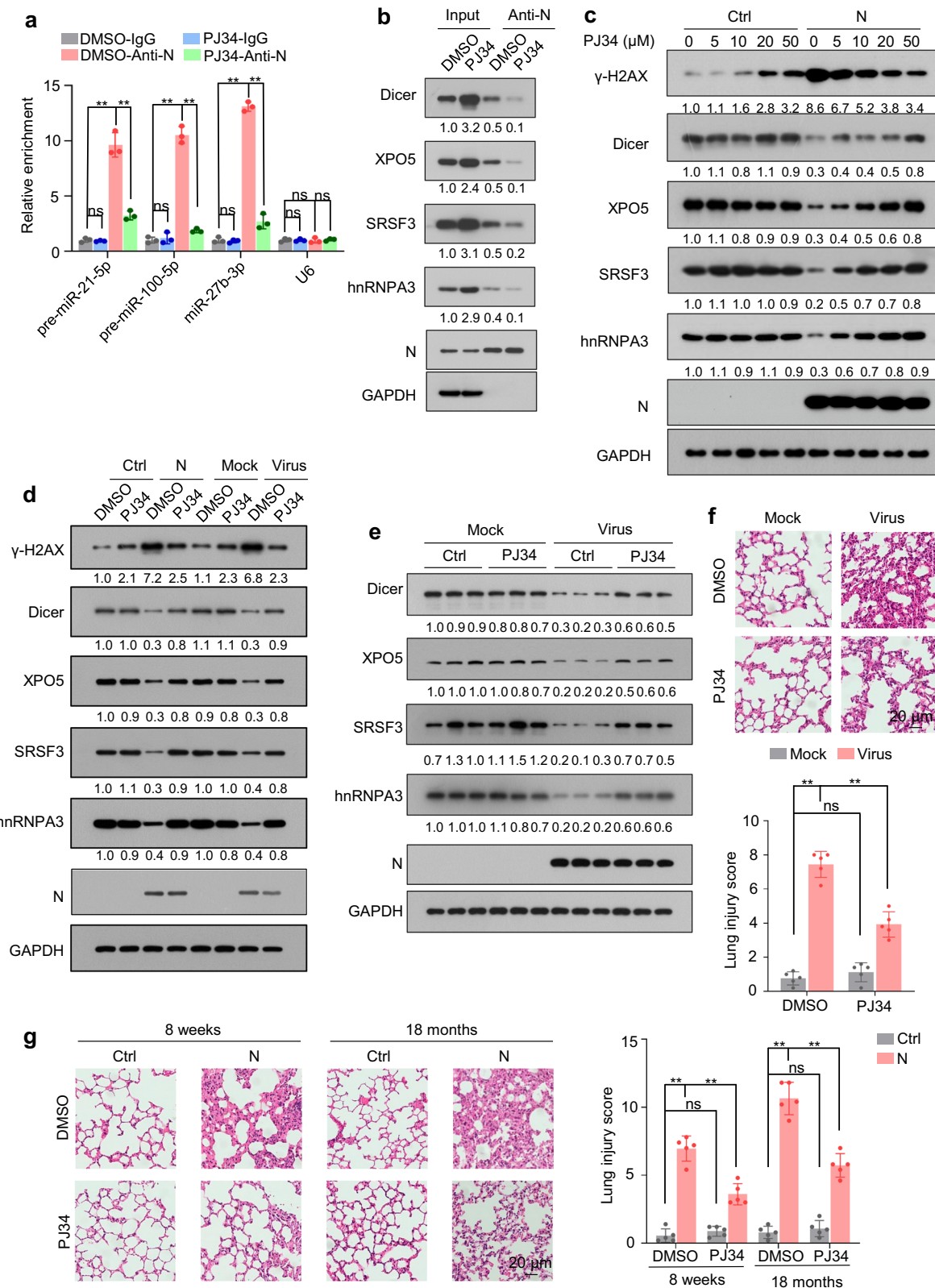

Subsequently, they were incubated with Protein A/G magnetic beads at 4 °C for 4 h. The beads were then washed five times with RIP washing buffer (50 mM Tris-HCl [pH 7.5], 150 mM KCl, 5 mM EDTA, 0.5% [v/v] NP-40, and 0.5 mM DTT) supplemented with 500 U/mL RNase inhibitor (Vazyme Biotech Co., Ltd.) and protease inhibitor cocktail tablets (Roche); RNA was extracted using the TRIzol reagent and quantified via RT-qPCR. Relative enrichment was calculated as

the ratio between specific antibodies and the normal IgG control, defined as 1.

**Protein synthesis measurement**

Nascent polypeptides were labeled by treating cells with 10 μg/mL puromycin (MedChemExpress) for 1 h, followed by immunoblotting with an anti-puromycin antibody (MABE343; Sigma-Aldrich, 1:5000).

**Fig. 9 | PJ34 relieves N protein-induced pneumonia.** See also Supplementary Fig. 9. **a**, **b** Lysates from A549-N cells treated with or without PJ34 (50 μM) for 2 h were subjected to immunoprecipitation using an anti-N protein antibody and then to RT-qPCR (**a**) and immunoblotting (**b**) to detect pre-miRNAs and proteins associated with N protein. **c** Immunoblotting of the indicated proteins in A549-Ctrl or A549-N cells after treatment with different concentrations of PJ34 for 2 h. **d** Immunoblotting of the indicated proteins in A549-Ctrl or A549-N cells (left) or A549-hACE2 cells infected with SARS-CoV-2 or mock (right) in the presence or absence of PJ34. **e**, **f** K18-hACE2 mice were intranasally infected with SARS-CoV-2 or mock-infected and treated with or without PJ34; lung tissues were subjected to immunoblotting with the indicated antibodies (**e**) and HE staining to assess lung injury (**f**). Representative immunoblots of lung tissues from three out of five mice per group are shown (**e**). **g** Mice (8-week-old and 18-month-old) were intranasally instilled with control plasmid or N protein-expressing plasmid and treated with or without PJ34; lung tissues were subjected to HE staining to assess lung injury. The numbers below the blots (**b**–**e**) indicate the relative densitometric quantification of the bands normalized to corresponding GAPDH bands. The mean values of three independent experiments are shown in (**b**–**d**). Data in (**a**) are expressed as mean ± SD of three independent replicates. Data in (**f**, **g**) are expressed as mean ± SD of five mice. **$p < 0.01$; ns not significant ($p > 0.05$; two-tailed unpaired Student's *t*-test). anti-N anti-N protein antibody, Ctrl control plasmid, N N protein. Source data and exact *p* values are provided in the Source Data file.

## Soluble and insoluble protein isolation

Ultracentrifugation was used to isolate soluble and insoluble proteins, as previously reported[76]. Briefly, cells were lysed using Triton X-100 buffer (1% [v/v] Triton X-100, 150 mM NaCl, 50 mM Tris-HCl [pH 8.0], and 1 mM EDTA) supplemented with protease inhibitor cocktail tablets (Roche) and sonicated at 4 °C for 1 min (10 s on and 10 s off) on a QSonica Q800R3 sonicator (QSonica). Protein concentration was quantified using a BCA Protein Assay Kit (Beyotime Biotechnology) to ensure that the starting amounts of protein in the samples were equal. Cell lysates were centrifuged at $100,000 \times g$ for 30 min using a Beckman TL-100 Ultracentrifuge (Beckman, Palo Alto, CA, USA) at 4 °C. The soluble proteins in the supernatant were collected for immunoblotting. The insoluble pellet was re-suspended in SDS lysis buffer (Triton X-100 buffer, 2% [m/v] SDS), supplemented with protease inhibitor cocktail tablets (Roche), and sonicated on a QSonica Q800R3 sonicator (QSonica) until the pellet became invisible. The BCA assay was performed to quantify the protein concentration in the pellet fraction. Soluble and insoluble protein samples were analyzed using immunoblotting.

## Confocal microscopy

**Immunofluorescence.** Immunofluorescence was used to detect cytosolic dsDNA, γ-H2AX foci, the R-loop, and colocalization of N protein with p62. Cells were fixed using 4% (m/v) paraformaldehyde diluted in PBS and incubated overnight with anti-dsDNA (Sigma-Aldrich; 1:500), anti-γ-H2AX (2577; Cell Signaling Technology, Danvers, MA, USA; 1:500), anti-DNA-RNA hybrid antibody (clone S9.6; MABE1095; Sigma-Aldrich; 1:500), anti-nucleocapsid protein (Sino Biological; 1:100), or anti-p62 (88588; Cell Signaling Technology; 1:200) antibodies at 4 °C. The cells were then incubated with goat anti-mouse-Alexa Fluor™ 488 (A-11001; Thermo Fisher Scientific; 1:1000) for detecting dsDNA, R-loop, and p62; with goat anti-rabbit-Alexa Fluor™ 488 (Thermo Fisher Scientific; 1:500) for detecting γ-H2AX; or with goat anti-rabbit-Alexa Fluor™ 555 (A-21428; Thermo Fisher Scientific; 1:500) for detecting the N protein at 37 °C for 30 min. Subsequently, the nuclei were counterstained with 200 ng/mL DAPI (Sigma-Aldrich) for 5 min, and cells were imaged using a confocal microscope (LEICA DMi8; Leica Microsystems). To calculate the average number of γ-H2AX and R-loop foci per cell, 60–80 cells were counted in an individual biological replicate, and the total number of cells in the three biological replicates was 200. The colocalization between N protein and p62 was analyzed using Pearson's correlation coefficient as previously reported[77]; 30–50 cells were analyzed in an individual biological replicate, and the total number of cells in the three biological replicates was 100.

**LC3 puncta imaging.** A549 cells were transduced with HBLV-EGFP-LC3-PURO lentivirus (Hanbio Biotechnology, Shanghai, China) and selected with 2 μg/mL puromycin (MedChemExpress). EGFP-LC3-expressing cells were transfected with an N protein-expressing plasmid or a control plasmid. The cells were fixed with 4% (m/v) paraformaldehyde 48 h post-transfection and incubated overnight with anti-nucleocapsid protein (Sino Biological; 1:100) at 4 °C. The cells were then incubated with goat anti-rabbit-Alexa Fluor™ 555 (A-21428;

Thermo Fisher Scientific; 1:500) to detect the N protein at 37 °C for 30 min. Subsequently, followed by counterstaining with 200 ng/mL DAPI (Sigma-Aldrich) for 5 min, and imaged using laser scanning confocal microscopy. To calculate the average number of EGFP-LC3-positive puncta per cell, 30–50 cells were counted in an individual biological replicate, and the total number of cells in the three biological replicates was 100.

**Proteotoxic stress analysis.** A549-Ctrl and A549-N cells transfected with pCI-neo Fluc-EGFP or pCI-neo FlucDM-EGFP constructs were fixed using 4% (m/v) paraformaldehyde 48 h post-transfection; A549-hACE2 cells transfected with pCI-neo Fluc-EGFP or pCI-neo FlucDM-EGFP constructs for 6 h and infected with SARS-CoV-2 or mock were fixed using 4% (m/v) paraformaldehyde 48 h post-infection, followed by counterstaining with 200 ng/mL DAPI (Sigma-Aldrich) for 5 min, and imaged using laser scanning confocal microscopy. To calculate the average number of Fluc-EGFP aggregates per cell, 60–80 cells were counted in an individual biological replicate and the total number of cells in the three biological replicates was 200.

## Gene Expression Omnibus (GEO) analysis

Transcriptome data were downloaded from the GEO database [https://www.ncbi.nlm.nih.gov/gds/]. The GSE178824 dataset comprised RNA sequencing data of monocytic myeloid-derived suppressor cells from patients with severe or asymptomatic COVID-19; GSE159585 comprised RNA sequencing data of lung tissues from deceased COVID-19 patients and individuals without COVID-19; GSE209891 comprised RNA sequencing data of lung tissues from mice of different ages. Differentially expressed genes were identified using the limma (version 3.54.2) R package[78]. Genes encoding components of the miRNA processing and RNA-splicing pathways were retrieved from the biological process categories in the Molecular Signatures Database [https://www.gsea-msigdb.org/gsea/msigdb]. The Z-score of each gene was calculated for each biological replicate and plotted and clustered using the heatmap function in R v 4.1.0 (R Foundation for Statistical Computing, Vienna, Austria) with hierarchical clustering based on the Euclidean distance.

## In silico protein-protein interaction analysis

Putative protein interactors of N protein were retrieved from BioGRID [https://thebiogrid.org/][42]. Pathway enrichment of putative N protein interactors was performed using the Metascape webserver [http://metascape.org][79].

## Small RNA-seq analysis

Small RNA sequencing was performed by Biomarker Technologies Corporation (Beijing, China). Clean sequencing reads were aligned to sequences hosted on the Silva, GtRNAdb, Rfam, and Repbase databases using Bowtie 1.0.0 [https://bowtie-bio.sourceforge.net/index.shtml] to filter small RNAs derived from ribosomal RNA, transfer RNA, small nuclear RNA, small nucleolar RNA, other noncoding RNAs, and repeat sequences. The remaining reads were aligned to sequences hosted on miRbase to identify known miRNAs. Differentially expressed miRNAs (|fold change| > 1.5 and p < 0.05) were identified using the

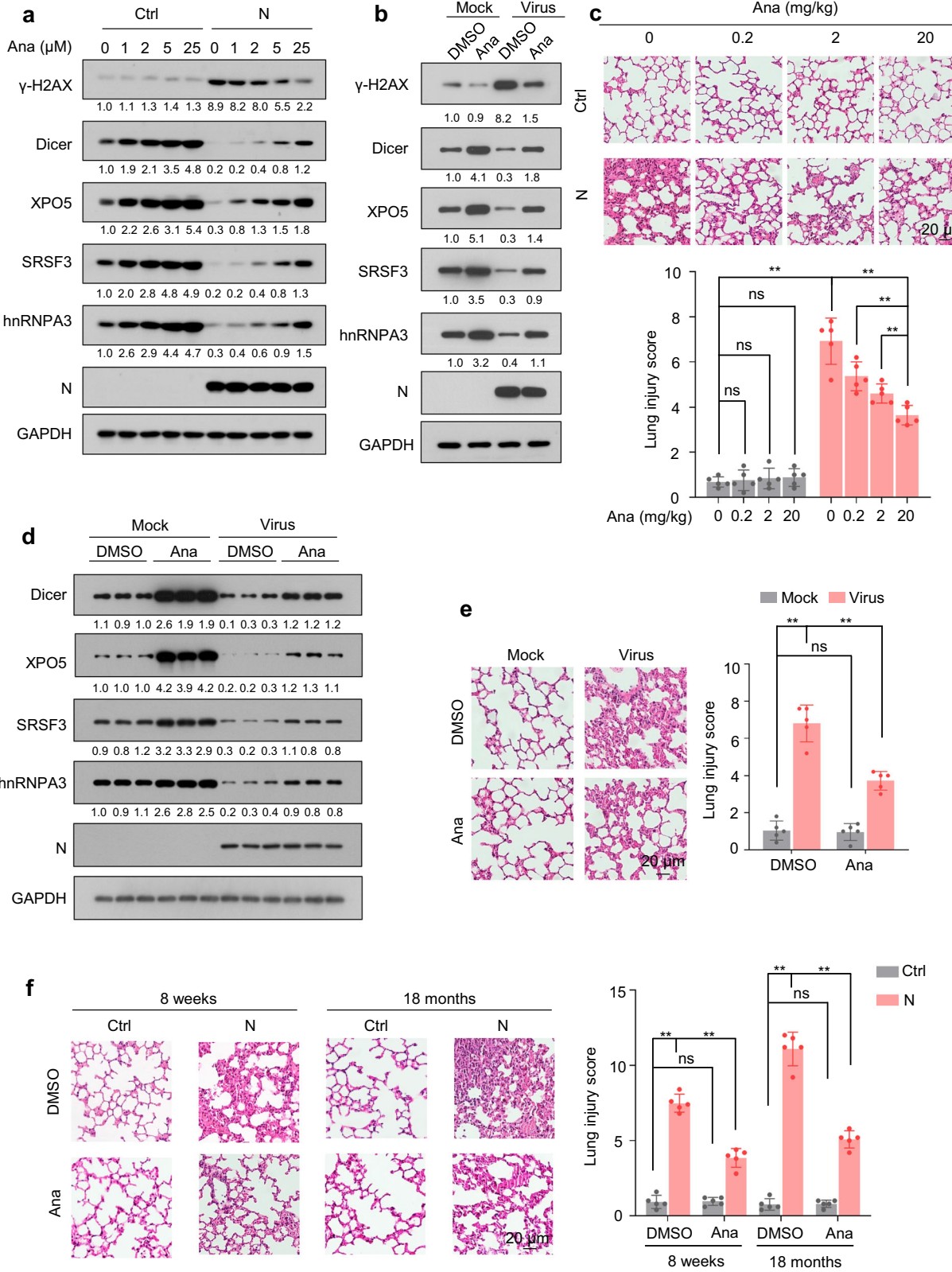

DESeq (version 1.10.1) package in R[80]. To generate a heatmap of the top 30 differentially expressed miRNAs with the highest expression, the Z-score of each miRNA was calculated for each biological replicate and then plotted and clustered using the heatmap function in R v 4.1.0 (R Foundation for Statistical Computing), with hierarchical clustering based on the Euclidean distance.

**Alternative splicing analysis**

Full-length transcriptome sequencing was performed by Biomarker Technologies Corporation using Oxford Nanopore Technology and run on the PromethION platform[81]. Alternative splicing events were analyzed using the PSI-Sigma method[82]. ΔPSI was calculated as follows: ΔPSI = (PSI value of A549-N cells) − (PSI value of A549-Ctrl cells). An

**Fig. 10 | Anastrozole relieves N protein-induced pneumonia.** See also Supplementary Fig. 10. **a** Immunoblotting of the indicated proteins in A549-Ctrl or A549-N cells treated with different concentrations of anastrozole for 24 h. **b** Immunoblotting of the indicated proteins in A549-hACE2 cells infected with SARS-CoV-2 or mock in the presence or absence of anastrozole. **c** Mice were intranasally instilled with control plasmid or N protein-expressing plasmid and treated with different doses of anastrozole as indicated; lung tissues were subjected to HE staining to assess lung injury. **d**, **e** K18-hACE2 mice were intranasally infected with SARS-CoV-2 or mock and treated with or without anastrozole (20 mg/kg); lung tissues were subjected to immunoblotting with the indicated antibodies (**d**) and HE staining to assess lung injury (**e**). Representative immunoblots of lung tissues from three out of five mice per group are shown (**d**). **f** Mice (8-week-old and 18-month-old) were intranasally instilled with control plasmid or N protein-expressing plasmid and treated with or without anastrozole (20 mg/kg); lung tissues were subjected to HE staining to assess lung injury. The numbers below the blots (**a**, **b**, **d**) indicate the relative densitometric quantification of the bands normalized to corresponding GAPDH bands. The mean values of three independent experiments are shown in (**a**, **b**). Data in (**c**, **e**, **f**) are expressed as mean ± SD of five mice. **$p < 0.01$; ns not significant ($p > 0.05$, two-tailed unpaired Student's $t$-test). anti-N anti-N protein antibody, Ctrl control plasmid, N N protein, Ana anastrozole. Source data and exact $p$ values are provided in the Source Data file.

absolute value of ΔPSI ≥ 10% was set as the cutoff for alternative splicing events with significant changes[83].

To analyze the overall intron retention (IR) levels in A549-Ctrl and A549-N cells, full-length transcriptome sequencing data were mapped to the human genome (hg19; known reference genes from the University of California, Santa Cruz [UCSC]) using Minimap2[84]. Exons were annotated using RefSeq, and introns were defined as the genomic regions not covered by any exons annotated using RefSeq[85]. The fraction of transcripts with IR over total sequencing reads was calculated as follows: Fraction of transcripts with IR = isoform reads containing introns ÷ total sequencing reads. The relative fraction of transcripts with IR in A549-Ctrl cells was defined as 1.

The intronic and exonic RNA levels of *GPX1*, *TERT*, and *CCN1* in A549-Ctrl or A549-N cells, BEAS-2B-Ctrl or BEAS-2B-N cells, and SARS-CoV-2- or mock-infected A549-hACE2 cells were determined via RT-qPCR. The relative ratio of intronic and exonic RNA levels was calculated and defined as 1 in Ctrl cells.

## Quantification and statistical analysis
Image quantification was performed using Fiji 2.3.0[86]. Data were presented as mean ± standard deviation of at least three independent experiments. Normal distribution was confirmed using the Shapiro–Wilk normality test, and homogeneity of variance was tested using Levene's test. Student's $t$-test was performed to compare the differences between two groups. In all experiments, the significance level was set as $α = 0.05$, and $p < 0.05$ indicated significant intergroup differences. Statistical analyses were performed using SPSS Statistics 23.0 (IBM SPSS Statistics, IBM Corporation, Armonk, NY, USA). Additional information regarding statistical tests, $p$ values, and sample size are described in Figure legends.

## Reporting summary
Further information on research design is available in the Nature Portfolio Reporting Summary linked to this article.

## Data availability
All data needed to evaluate the conclusions are included in the paper and/or the Supplementary Information and/or the Source Data file. The small RNA data generated in this study have been deposited in the NCBI Sequence Read Archive database under BioSample accession code PRJNA1000280. The full-length transcriptome sequencing data generated in this study have been deposited in the NCBI Sequence Read Archive database under BioSample accession code PRJNA1000513. Source data are provided with the paper. Source data are provided with this paper.

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

## Acknowledgements

We thank Dr. Yan Wu and Fangxu Li (Wuhan Institute of Virology of Chinese Academy of Sciences) and Ruixue Liu, Dr. Yunpeng Liu, Ruimin Zhu, and Dr. Xintian Zhang (Beijing Key Laboratory for Animal Models of Emerging and Reemerging Infectious Diseases, National Center of Technology Innovation for Animal Models, Institute of Laboratory Animal Sciences, CAMS & PUMC) for their assistance with SARS-CoV-2 virus preparation and infection. We would like to thank Prof. Ni Tang, Prof. Xuemei Lian, Prof. Yong Lin, Dr. Dan Shi, Dr. Jie Hu (Chongqing Medical University) and Prof. Yong Peng (Sichuan University) for providing cell lines and plasmids. We also wish to thank Hang Liang, Junnan Liu, Meimei Shen, Bingxin Tang and Linxin Zheng (Chongqing Medical University) and Ziheng Tang (Chongqing Nankai Middle School) for performing bioinformatic analysis, cell culture, western blotting and other experiments. This work was supported by the National Natural Science Foundation of China (grant numbers 82172915 to K.F.T., 81972648 to K.F.T., and 81773011 to K.F.T.), National Key Research and Development Program (grant number 2023YFC3041500 to X.P.), Chongqing Medical University Program for Youth Innovation in Future Medicine (grant number W0084 to K.F.T.), Science and Technology Innovation Project of Chongqing Medical University (to K.F.T.), and Chongqing Postdoctoral Science Foundation (grant number CSTB2023NSCQ-BHX0134 to Y.W.L.).

## Author contributions

Conceptualization: K.F.T.; Methodology: K.F.T., Y.W.L.; Investigation: Y.W.L., J.P.Z., H.J., D.X., A.Z., X.W., Z.D., Z.L., F.C., X.Y.W., Y.B., D.C., Y.C., Q.W., Y.Y., X.Z., K.F.T.; Validation: Y.W.L., J.P.Z., H.J., D.X., A.Z., X.W., Z.D., Z.L., F.C., X.Y.W., Y.B., D.C., K.F.T.; Visualization: K.F.T., Y.W.L., J.P.Z., H.J.; Data curation: K.F.T., Y.W.L., J.P.Z., H.J.; Resources: K.F.T., X.P., D.X., X.Z., S.C., Funding acquisition: K.F.T., Y.W.L., X.P.; Project administration: K.F.T., Y.W.L.; Supervision: K.F.T., A.L.H., X.P.; Writing—original draft: K.F.T., Y.W.L.; Writing—review & editing: K.F.T.

## Competing interests

The authors declare no competing interests.
