## [Peer Review File · Nature Communications]

SARS-CoV-2 N protein-induced Dicer, XPO5, SRSF3, and hnRNPA3 downregulation causes pneumoniaREVIEWER COMMENTS

Reviewer #1 (Remarks to the Author):

General comments.

The study by Yu-Wei Luo and colleagues investigates the impact and consequences of SARS-CoV-2 N on fundamental host cellular pathways, such as RNA interference (RNAi) and mRNA splicing. Mechanistically, the authors show that the viral N-protein triggers the autophagic degradation of DICER and XPO5 – key components that regulate miRNA biogenesis and functions – as well as SRSF3 and hnRNPA3, both involved in mRNA splicing. The loss of such cellular factors, in turn, causes DNA damage and hampers protein homeostasis, ultimately leading to pneumonia in vivo in murine models. Importantly, the authors demonstrate that the pharmacological rescue of DICER, XPO5, SRSF3 and hnRNPA3 expression alleviates lung injury in SARS-CoV-2 N-expressing mice. The present study also highlights a correlation between ageing, the reduced expression of RNAi and mRNA splicing factors, and the severity of COVID-19 symptoms.

The work is well-conceived and designed. The data set is comprehensive and robust, and authors' claims are supported by the obtained results. I have a few concerns that I encourage the authors to address to improve their manuscript, thus making it suitable for publication in Nature Communications.

Limitation of the study:

1) One important limitation of the study is that N protein overexpression is compared to control-non overexpressing cells. Although initial overexpression of different viral protein is compared in inducing RNAi defect and alteration in slicing of an intron-containing luciferase reporter minigene, and N-protein is the more active, I believe many of the in vivo experiments lack this important control. I would appreciate to see that overexpression of any other viral factors do not induce the same proteotoxic stress and autophagy deregulation.

Indeed N-protein itself can be insoluble. The experiment in figure 4a shows that N protein increases the fraction of insoluble proteins. Is this N-protein specific? Is the insoluble fraction N-protein itself? Does it occur with any overexpressed RNA binding viral protein?

I believe also this experiment lacks a comparison with other overexpressed viral proteins.

Indeed N-protein associate with p62 suggesting that the autophagic pathway is strongly activated by N-protein, as it normally occurs for insoluble proteins. Indeed p62 KD do not impact DICER level in cells not expressing N-protein. My impression is that any overexpressed protein, especially is partially insoluble and prone to undergo LLPS, might do the same.

2) In addition, the authors suggest that DICER, XPO5 downregulation induces more protein synthesis and proteotoxic stress via the JNK pathway. This is in line with the evidences that several

microRNA are lost in cells expressing N-protein. One important mediator of microRNA dependent translation inhibition are the GW182-like factors (TNRC6 A, B and C). Are these factors also affected by N-protein expression?

Specific remarks.

One issue of the present study is that it partially fails to quote the relevant literature regarding the interplay between SARS-CoV-2 N protein and DNA damage, and between RNAi components and DNA damage signaling and repair.

For instance, it has been recently shown that SARS-CoV-2 N causes DNA damage accumulation by interfering with the recruitment at damaged sites of 53BP1, an important factor involved in DNA damage repair (10.1038/s41556-023-01096-x). In addition, 53BP1 recruitment at sites of DNA double-strand breaks (DSBs) also depends on DICER activity (10.1038/nature11179; 10.1242/jcs.182188; 10.1083/jcb.201612131; 10.1038/s41598-019-42892-6), which it has been shown here to be inhibited by the N-protein (Figure 1). Besides quoting these studies, the authors may also want to address if the ability of 53BP1 to localize at DSBs, which is reduced in cells expressing SARS-CoV-2 N, is rescued by anastrozole administration. This would strengthen their findings on the effects of SARS-CoV-2 N on DNA repair.

Regarding the use of anastrozole, the authors have previously shown and published that this molecule is able to increase DICER expression (10.7150/THNO.41894). However, and unexpectedly, this compound also seems to stimulate the expression of the other factors that are found to be downregulated by SARS-CoV-2 N (i.e., XPO5, SRSF3 and hnRNPA3; Figure 7e). This suggests that anastrozole could function at a more upstream level, for instance, as an inhibitor of the autophagic pathway, in a way similar to chloroquine (Figure 1e). The authors should test such possibility or comment on this.

Reviewer #2 (Remarks to the Author):

Luo and colleagues present a study on the effects of SARS-CoV-2 Nucleocapsid protein on the RNAi pathway and alterations to RNA splicing and proteostasis, common features of aging.

Mechanistically, the authors attribute N-induced degradation of Dicer, XPO5, SRSF3, hnRNPA3 to compromised RNA splicing, RNAi and increased R-loop formation, which in turn is linked to increases in DNA damage and proteotoxic stress. In mouse models, this N protein-driven damage caused pneumonia in young mice and led to more severe pneumonia in older mice. The authors present biochemical data to show that SARS-CoV-2 N protein interacts with components of these pathways (Dicer, XPO5...) and leads to their degradation in an RNA-dependent manner. The authors also provide data indicating that N induces DNA damage and represses miRNA production and RNA splicing, leading to increased translation of insoluble intron-containing proteins contributing to proteotoxic stress. Further, they provide data to indicate that N protein expression is sufficient to induce pneumonia in mice, and that pneumonia triggered in older mice is more severe compared to

N-induced pneumonia in younger mice. In addition, the authors reveal efficacy of two cancer drugs in reversing N protein-induced pneumonia in the ectopic expression mouse model.

The data presented in this paper are compelling but overall the study suffers from lack of careful validation analysis in vivo in mouse models in parallel with the studies of over-expressed N protein. Moreover, the SARS2 infection studies in vitro, shown in the supplemental figures, should be move to main figures to show infection phenotype validation of the N over expression phenotype.

The abstract and introduction present the study as one focused on the biology of aging and its impact on SARS-CoV-2 induced pneumonia severity, however the paper's main questions are more mechanistically focused on the actions of the N protein on host homeostasis, which is then linked to pneumonia severity in young and older mice. Restructuring the abstract and introduction to focus on N-driven mechanisms that are investigated would increase the pertinence of the abstract and introduction sections.

While the authors propose clinical applications for the use of PJ34 and anastrozole, it would be valuable to include evidence that anastrozole improves the age-driven phenotype of worse COVID-19/N-induced pneumonia in older mice. In addition, expanding the discussion of roles the aromatase inhibitor anastrozole may play in regulating TMPRSS2 may be relevant as this has been suggested in the literature. Given the presented evidence that the inhibitor PJ34 induces DNA damage in uninfected cells, some discussion on the feasibility of its use as a COVID-19 treatment would be beneficial. Including a direct comparison (on the same immunoblot) of N protein expression in the overexpression systems (both plasmid overexpression and stable cell line expression) and viral infection would be helpful to draw parallels between the two systems and bolster the relevance of findings within an infection context.

Figure 1: What were the basis for selecting the specific SARS2 proteins for the siRNA and splicing regulation screen? Figs 1C-e, and figure S2d need quantification to show the magnitude of the regulation or protein levels. Fig 1j, the p62 knockdown partial relief of N protein-induced downregulation of Dicer, XPO5, 159 SRSF3, and hnRNPA3 proteins is interesting to suggest that N is regulating a p62 scaffold protein complex. More information is needed here to define this complex and how N protein interacts with it during SARS2 infection. Moreover, the N protein interactions with p62, Dicer, XPO5, SRSF3160 and hnRNPA3 proteins need to be validated in a true SARS2 infection.

Figure 1A and S2B. Need a protein expression negative control nonbinding to N for comparative analyses

Figure 1e. and Supplementary figure 2d. While these are different blots, the notable difference in phenotype seen between the DMSO control treatments in Figure 1e and Figure 2d is concerning, given that these should recapitulate one another. There is markedly less N-induced degradation of Dicer and other proteins in the supplementary Figure 2d DMSO control compared to Figure 1e, which calls into question the data in Figure 1e.

For experiments pointing to autophagy-mediated degradation of these proteins, it would be useful to include additional autophagy inhibitors given their more direct mechanisms of action compared to Chloroquine. This is especially important given that Chloroquine induces extensive disorganization of Golgi and endo-lysosomal systems whereas other available inhibitors are more

targeted in their block of lysosome acidification or PI3K complex and early phagosome formation.

Figure 1f. Including immunofluorescence staining of N protein is important here to show transfection efficiency and link the LC3 phenotype to N protein expression. Including more detailed descriptions on the methodology used for analysis and more information about how many cells were quantified in each experiment would be beneficial as well, as it is unclear what the 200 cell number refers to.

Figure 1i. Missing analysis of colocalization between p62 and N—the representative image shown is not convincing and formal analysis of colocalization is needed.

Figure 1h. The presented interaction between N protein and p62 is not robust. Additionally, the IgG control looks as though there is non-specific interactions especially for the Dicer blot.

Figure 2. Panel A needs quantification to show the phosphorylation induction level of each protein. Blots in Fig S3 need full quantification to show the differences in protein abundance.

The analyses shown in Figs 3 and S4 are compelling but need to be validated in an actual SARS2 infection model, which should be moved from the supplemental figs to these main figs. . Fig 3d-f miRNA nuclear retention assessment lacks important specificity controls to include mRNA and rRNA controls to show specificity of N for retention of miRNA over these other RNAs in the nucleus. rRNA should be included as an internal control for both nuclear and cytosolic RNAs. Figs. 3i-m needs validation of these outcomes in actual SARS2 infected cells- move to main figs.

Figure 4. Unclear about the conclusions regarding protein synthesis v.s protein translation as impacted by N protein given that N protein appears to increase both protein translation and synthesis in the data presented, however description of these same data in the results section contradicts this by stating that N increases translation while inhibiting protein synthesis, which is further contradicted in Supplementary Figure 7. In panel h of Figure 4, reporting densitometry values would be helpful given very subtle differences in blots. Fig 4A needs quantification. This figure shows increased soluble proteins when N is expressed in addition to increased insoluble proteins. This relationship needs to be explained and examined for how N is inducing a general protein upregulation. 4d needs band quantification to show the true differences that the authors are eluding to. Figs 4f-l need quantification.

As noted above, the experiments and outcomes in Fig 1-4 all need validation in SARS2 infection models such as in the A549 or Beas2B cells used in these studies. This is a critical omission of the larger study where the N protein regulation of dicer and factors needs to be shown in the context of a real SARS2 infection, and hence the infection data sets should be moved to main figures.

Fig. 5-7 blots: the bands need to be quantified for protein abundance measurements.

The experiments in figures 5-7 need to be repeated in mice undergoing SARS2 infection. Since the

delta variant SARS2 emergence, the virus is capable of infecting B6 mice, for example or the use a mouse-adapted virus can be applied to do in vivo infection and validation analyses of the fig 5-7 data sets.

Figure 7. The dosage used for anastrozole in mice (20mg/kg) is far higher than what is used in human patients for breast cancer (1 mg/day)—given possible safety concerns about such high doses, it may be important to discuss the viability of using this as a treatment in humans. How old are the mice used in these experiments? It is important to supply evidence that anastrozole reverses the age-related phenotype of worse pneumonia in older individuals beyond just the N-driven changes in younger mice. This would be an especially valuable experiment to include if proposing to use this drug as a treatment for human COVID-19.

Supplementary Figures 1, 2. Having protein-level data to validate the transcript data in figure 1 (a,b) would be valuable, especially given that later in the paper, in Supplementary Figure 2 (panel g) it is stated that SARS-CoV-2 infection did not affect Dicer, XPO5, SRSF3, hnRNPA3 at the mRNA level and infection induces changes only at the protein level. This calls into question what the heatmaps in Supplementary Figure 1a, 1b are showing and if this data is relevant or contradicting the experimental findings of this paper.

Reviewer #3 (Remarks to the Author):

This manuscript explores the role of the N protein of SARS-CoV-2 in affecting the levels of key proteins of the RNAi pathway and the splicing machinery. The authors show that N expression translates into decreased protein expression, leading to proteotoxic stress and DNA damage. Because the factors targeted by N have decreased levels in aged mice, the authors suggest that N activity towards such factors could contribute to the severity the symptoms in aged individuals. They finish by testing two compounds that could be promising in alleviating N-driven degradation. The manuscript is rather data-heavy, well written and pleasant to read. Appart from data presentation suggestions (detailed after) the experiments are well-controlled and interpreted. It is nice work, but I have a major concern with the experimental setup. The authors perform all their experiments with plasmid-driven N overexpression. During an infection with SARS-CoV-2, the N protein will be loaded on viral RNA, therefore unable to massively bind cellular RNAs. Because the interaction of N with RNAi and splicing factors depends on RNA, one could expect that it would not be as good of a binder of such cellular factors. The authors need to demonstrate that what they see with plasmid-driven overexpression occurs during viral infection. It would be unrealistic to repeat the entire paper with virus, but key experiments should be performed with actual infection (including in vivo experiments).

Major comments

– all the Western blots and microscopy images need to be properly quantified and differences

represented as bar graphs. The proper statistical tests need to be performed.

- what is the control for mouse experiments? The authors should express an unrelated protein and not only infuse mice with a plasmid, to control for the effect of plasmid-driven protein production.
- If anastrozole acts by enhancing Dicer expression, how come that it also affects XPO5 and splicing factors expression?

Minor comments

- Is the interaction of N with p62 RNA-dependent?
- gene expression maps: please indicate names of the genes of interest, for example DICER in Suppl Fig 1a. For Fig 3a, please label with the miRNA names.

REVIEWER COMMENTS

Reviewer #1 (Remarks to the Author):

General comments.

The study by Yu-Wei Luo and colleagues investigates the impact and consequences of SARS-CoV-2 N on fundamental host cellular pathways, such as RNA interference (RNAi) and mRNA splicing. Mechanistically, the authors show that the viral N-protein triggers the autophagic degradation of DICER and XPO5 – key components that regulate miRNA biogenesis and functions – as well as SRSF3 and hnRNPA3, both involved in mRNA splicing. The loss of such cellular factors, in turn, causes DNA damage and hampers protein homeostasis, ultimately leading to pneumonia in vivo in murine models. Importantly, the authors demonstrate that the pharmacological rescue of DICER, XPO5, SRSF3 and hnRNPA3 expression alleviates lung injury in SARS-CoV-2 N-expressing mice. The present study also highlights a correlation between ageing, the reduced expression of RNAi and mRNA splicing factors, and the severity of COVID-19 symptoms.

The work is well-conceived and designed. The data set is comprehensive and robust, and authors' claims are supported by the obtained results. I have a few concerns that I encourage the authors to address to improve their manuscript, thus making it suitable for publication in Nature Communications.

Response: We thank the reviewer for the careful review of our manuscript and encouraging comments. We have performed additional experiments as suggested and addressed all questions and comments in the revised manuscript.

Limitation of the study:

1) One important limitation of the study is that N protein overexpression is compared to control-non overexpressing cells. Although initial overexpression of different viral protein is compared in inducing RNAi defect and alteration in slicing of an intron-containing luciferase reporter minigene, and N-protein is the more active, I believe many of the in vivo experiments lack this important control. I would appreciate to see that overexpression of any other viral factors do not induce the same proteotoxic stress and autophagy deregulation.

Response: Thank you for this constructive suggestion. We conducted additional experiments to investigate the effects of various SARS-CoV-2 proteins, including spike, membrane, envelope, and NSP8, on DNA damage, proteotoxic stress, protein synthesis, lung injury, and the expression of Dicer, XPO5, SRSF3, and hnRNPA3. The results are presented in Supplementary Fig. 2f, 2g, 3d, 3e, 6b, 6c, 6d, 6f, and 7d in the revised manuscript. Specifically, we found that

membrane and envelope proteins did not induce DNA damage (Supplementary Fig. 3d and 3e). Moreover, ectopic expression of spike, membrane, and NSP8 proteins did not increase protein synthesis or induce proteotoxic stress; the ectopic expression of envelope protein even repressed protein synthesis (Supplementary Fig. 6b, 6c, 6d, 6f). NSP8 expression in mouse lung tissues did not induce lung injury (Supplementary Fig. 7d). The ectopic expression of spike, membrane, and envelope proteins did not trigger autophagic degradation of Dicer, XPO5, SRSF3, and hnRNPA3 proteins (Supplementary Fig. 2f).

Indeed N-protein itself can be insoluble. The experiment in figure 4a shows that N protein increases the fraction of insoluble proteins. Is this N-protein specific? Is the insoluble fraction N-protein itself? Does it occur with any overexpressed RNA binding viral protein?

I believe also this experiment lacks a comparison with other overexpressed viral proteins.

Response: Thank you for these comments. We found that N protein was mainly detected in the soluble fraction; only a very small amount of N protein was detected in the insoluble fraction (Supplementary Fig. 6a).

In contrast to that of N protein, the ectopic expression of spike, membrane, and envelope proteins did not increase the synthesis of insoluble proteins (Supplementary Fig. 6c). Moreover, we found that NSP8, an RNA-binding viral protein, did not seem to affect the synthesis of insoluble proteins (Supplementary Fig. 6b). These findings suggest that N protein, not other viral proteins, including RNA-binding viral proteins, enhances the synthesis of insoluble proteins.

Indeed N-protein associate with p62 suggesting that the autophagic pathway is strongly activated by N-protein, as it normally occurs for insoluble proteins. Indeed p62 KD do not impact DICER level in cells not expressing N-protein. My impression is that any overexpressed protein, especially is partially insoluble and prone to undergo LLPS, might do the same.

Response: Thank you for this comment. It is well known that protein aggregates can activate autophagy and induce their degradation. Previous reports have shown that N protein can condense with RNA through liquid-liquid phase separation (PMID: 33200826,33782395,33290746). However, we observed that, under our experimental conditions, N protein was mostly soluble (Supplementary Fig. 6a). Therefore, it is unlikely that N protein induces autophagy due to the formation of insoluble aggregates. Further studies are needed to elucidate how N protein induces autophagy. However, this is out of the scope of the current manuscript.

2) In addition, the authors suggest that DICER, XPO5 downregulation induces more protein synthesis and proteotoxic stress via the JNK pathway. This is in line with the evidences that

several microRNA are lost in cells expressing N-protein. One important mediator of microRNA dependent translation inhibition are the GW182-like factors (TNRC6 A, B and C). Are these factors also affected by N-protein expression?

Response: Thank you for this question. We conducted additional experiments and found that N protein did not influence the expression of TNRC6A, TNRC6B, or TNRC6C. As these findings are not crucial for the current manuscript, they have not been included in the revised manuscript. We present these data below for your assessment.

Figure 1 for reviewers only. TNRC6A, TNRC6B, and TNRC6C levels are not affected by N protein. A549-Ctrl and A549-N cells were subjected to immunoblotting. GAPDH was used as an internal control. The numbers below the blots represent the densitometric quantification of the bands normalized to the GAPDH band.

Specific remarks.

One issue of the present study is that it partially fails to quote the relevant literature regarding the interplay between SARS-CoV-2 N protein and DNA damage, and between RNAi components and DNA damage signaling and repair.

For instance, it has been recently shown that SARS-CoV-2 N causes DNA damage accumulation by interfering with the recruitment at damaged sites of 53BP1, an important factor involved in DNA damage repair (10.1038/s41556-023-01096-x). In addition, 53BP1 recruitment at sites of DNA double-strand breaks (DSBs) also depends on DICER activity (10.1038/nature11179; 10.1242/jcs.182188; 10.1083/jcb.201612131; 10.1038/s41598-019-42892-6), which it has been shown here to be inhibited by the N-protein (Figure 1). Besides quoting these studies, the authors may also want to address if the ability of 53BP1 to localize at DSBs, which is reduced in cells expressing SARS-CoV-2 N, is rescued by anastrozole administration. This would strengthen their findings on the effects of SARS-CoV-2 N on DNA repair.

Response: Thank you for this suggestion. These studies have been cited and discussed in the revised manuscript.

Consistent with a previous report that N protein impairs recruitment of 53BP1 at DSBs and inhibits non-homologous end-joining (NHEJ) (PMID: 36894671), we found that ectopic expression of N protein reduced the efficiency of NHEJ, but this effect was partially reversed by anastrozole. As Dicer is essential for 53BP1 recruitment at DSBs (PMID: 22722852, 26906421, 28642363), our findings may explain why N protein impairs 53BP1 recruitment at DSBs, hindering NHEJ.

Since these data merely replicate previously published findings, they are not essential for the current manuscript. Additionally, due to the huge amount of data in this manuscript, there is insufficient space to include these data in the revised manuscript. Therefore, these data have not been incorporated into the revised manuscript but are presented below for your evaluation.

Figure 2 for reviewers only. N protein inhibits NHEJ efficiency, which is partially rescued by anastrozole. Proportions of EGFP-positive cells were determined 48 h after co-transfection of A549-I-SceI-Hygro-EGFP cells with the I-SceI-expressing plasmid (pCBASceI) and N protein-expressing or control plasmids, and subsequent treatment of cells with or without anastrozole. Ctrl, control plasmid; N, N protein; Ana, anastrozole. Please refer to our previously published paper for detailed methods (PMID: 27685628).

Regarding the use of anastrozole, the authors have previously shown and published that this molecule is able to increase DICER expression (10.7150/THNO.41894). However, and unexpectedly, this compound also seems to stimulate the expression of the other factors that are found to be downregulated by SARS-CoV-2 N (i.e., XPO5, SRSF3 and hnRNPA3; Figure 7e). This suggests that anastrozole could function at a more upstream level, for instance, as an inhibitor of the autophagic pathway, in a way similar to chloroquine (Figure 1e). The authors should test such possibility or comment on this.

Response: Thank you for this interesting comment. Previous studies have shown that anastrozole does not stimulate autophagy (PMID: 37749673, 34391846). We found that anastrozole affected neither LC3-II nor p62 levels in both N protein-expressing cells and cells not expressing N protein. These findings exclude the possibility that anastrozole increases the expression of Dicer, XPO5, SRSF3, and hnRNPA3 by preventing autophagic degradation. However, as these results are not essential for the current manuscript, they have not been included in the revised manuscript. Therefore, we present these data below for your evaluation. Although the mechanisms underlying the induction of Dicer, XPO5, SRSF3, and hnRNPA3 expression by anastrozole remain to be elucidated, this is out of the scope of the current study and will be investigated in the future.

Figure 3 for reviewers only. Effects of anastrozole on autophagy. LC3-II and p62 levels in A549-Ctrl and A549-N cells treated with or without anastrozole (5 μ M) for 24 h. The numbers below the blots represent the densitometric quantification of the bands normalized to the corresponding GAPDH band; the mean values of three independent experiments are shown. Ctrl: control plasmid; N: N protein; Ana: anastrozole.

Reviewer #2 (Remarks to the Author):

Luo and colleagues present a study on the effects of SARS-CoV-2 Nucleocapsid protein on the RNAi pathway and alterations to RNA splicing and proteostasis, common features of aging.

Mechanistically, the authors attribute N-induced degradation of Dicer, XPO5, SRSF3, hnRNPA3 to compromised RNA splicing, RNAi and increased R-loop formation, which in turn is linked to increases in DNA damage and proteotoxic stress. In mouse models, this N protein-driven damage caused pneumonia in young mice and led to more severe pneumonia in older mice. The authors present biochemical data to show that SARS-CoV-2 N protein interacts with components of these pathways (Dicer, XPO5...) and leads to their degradation in an RNA-dependent manner. The authors also provide data indicating that N induces DNA damage and represses miRNA production and RNA splicing, leading to increased translation of insoluble intron-containing proteins contributing to proteotoxic stress. Further, they provide data to indicate that N protein expression is sufficient to induce pneumonia in mice, and that pneumonia triggered in older mice is more severe compared to N-induced pneumonia in younger mice. In addition, the authors reveal efficacy of two cancer drugs in reversing N protein-induced pneumonia in the ectopic expression mouse model.

The data presented in this paper are compelling but overall the study suffers from lack of careful validation analysis *in vivo* in mouse models in parallel with the studies of over-expressed N protein. Moreover, the SARS2 infection studies *in vitro*, shown in the supplemental figures, should be moved to main figures to show infection phenotype validation of the N over expression phenotype.

Response: Thank you for the careful review of our manuscript and your encouraging and constructive comments. We have performed additional experiments using real virus to substantiate our findings based on N protein overexpression both *in vivo* and *in vitro*. The results are presented in the following figures in the revised manuscript: Fig. 1a, 1b, 1f, 1j, 1i, 2d, 2f, 3d, 3e, 3j, 4c, 4e, 4g, 4i, 4k, 4m, 5c, 5g, 5i, 6b, 6e, 6g, 7j, 7k, 9d, 9e, 9f, 10b, 10d, 10e, as well as Supplementary Fig. 2e, 4f, 7g. Additionally, we have moved the *in vitro* SARS-CoV-2 infection data from the supplementary figures to the main figures.

The abstract and introduction present the study as one focused on the biology of aging and its impact on SARS-CoV-2 induced pneumonia severity, however the paper's main questions are more mechanistically focused on the actions of the N protein on host homeostasis, which is then linked to pneumonia severity in young and older mice. Restructuring the abstract and introduction to focus on N-driven mechanisms that are investigated would increase the pertinence of the abstract and introduction sections.

Response: Thank you for this valuable feedback. We have restructured the abstract and introduction according to your advice.

While the authors propose clinical applications for the use of PJ34 and anastrozole, it would be valuable to include evidence that anastrozole improves the age-driven phenotype of worse COVID-19/N-induced pneumonia in older mice. In addition, expanding the discussion of roles the aromatase inhibitor anastrozole may play in regulating TMPRSS2 may be relevant as this has been suggested in the literature.

Response: Thank you for these suggestions; we conducted additional experiments to compare the effects of PJ34 and anastrozole on N protein-induced pneumonia in young (8-week-old) and old (18-month-old) mice. Our results showed that PJ34 and anastrozole reversed the N-induced decrease in Dicer, XPO5, SRSF3, and hnRNPA3 levels, improving N protein-induced pneumonia in both 8-week-old and 18-month-old mice. The data are shown in Fig. 9g and 10f, as well as Supplementary Fig. 9e and 10e.

Moreover, we found that PJ34 and anastrozole increased the expression of Dicer, XPO5, SRSF3, and hnRNPA3 in SARS-CoV-2-infected mouse lung tissues and ameliorated pneumonia following SARS-CoV-2 infection. These data are presented in Fig. 9e, 9f, 10d, and 10e in the revised manuscript.

Anastrozole reportedly inhibits the conversion of testosterone into estradiol (PMID: 24451543). As testosterone can increase, while estradiol can repress TMPRSS2 expression (PMID: 34210968, 35103351), whether anastrozole can promote SARS-CoV-2 infection by upregulating TMPRSS2 expression requires further investigation. We have discussed this issue in the revised manuscript.

Given the presented evidence that the inhibitor PJ34 induces DNA damage in uninfected cells, some discussion on the feasibility of its use as a COVID-19 treatment would be beneficial. Including a direct comparison (on the same immunoblot) of N protein expression in the overexpression systems (both plasmid overexpression and stable cell line expression) and viral infection would be helpful to draw parallels between the two systems and bolster the relevance of findings within an infection context.

Response: Thank you for these suggestions. Given that PJ34 can induce DNA damage in cells not expressing N protein, further studies are required to determine if PJ34 can be used to treat SARS-CoV-2-induced pneumonia. This issue has been discussed in the revised manuscript.

We conducted additional experiments and found that the level of N protein expression in plasmid-overexpressing cells is comparable to that in SARS-CoV-2-infected cells. These results are presented in Fig. 9d of the revised manuscript.

Figure 1: What were the basis for selecting the specific SARS2 proteins for the siRNA and splicing regulation screen?

Response: We are interested in the four structural proteins encoded by SARS-CoV-2, which are usually targets for antigen tests or vaccines against COVID-19 (PMID: 38376212, 33340022). Therefore, in Fig. 1, we selected these proteins for investigation.

Figs 1C-e, and figure S2d need quantification to show the magnitude of the regulation or protein levels. Fig 1j, the p62 knockdown partial relief of N protein-induced downregulation of Dicer, XPO5, 159 SRSF3, and hnRNPA3 proteins is interesting to suggest that N is regulating a p62 scaffold protein complex. More information is needed here to define this complex and how N protein interacts with it during SARS2 infection. Moreover, the N protein interactions with p62, Dicer, XPO5, SRSF3 and hnRNPA3 proteins need to be validated in a true SARS2 infection.

Response: We have quantified all the western blots in the revised manuscript.

p62 is an autophagy receptor protein that recruits specific cargoes to phagophores. Our findings suggest that via interacting with N protein, p62 recruits the N protein-interacting proteins, including Dicer, XPO5, SRSF3, and hnRNPA3, to phagophores and induces their autophagic degradation.

Moreover, interactions between N protein and p62, Dicer, XPO5, SRSF, and hnRNPA3 proteins were validated in SARS-CoV-2 infected cells. These results are presented in Fig. 1f and 2f of the revised manuscript. During the viral infection, interaction between viral proteins within virus particles and cellular proteins are commonly observed. Therefore, it is expected that N protein can interact with cellular proteins in SARS-CoV-2-infected cells. Upon infection, the virus must undergo uncoating before viral RNA replication, transcription, and translation (PMID: 33116300, 38262893). The N protein may be released from the virion during uncoating and interact with cellular proteins. Additionally, before virus assembly, the newly synthesized N protein may also bind with cellular proteins.

Figure 1A and S2B. Need a protein expression negative control nonbinding to N for comparative analyses

Response: We found that GAPDH cannot bind to the N protein and thus used it as a negative control for the Co-IP assays. Please refer to Fig. 1e, 1f, 1h, 2e, 2f, and 9b and Supplementary Fig. 2a in the revised manuscript.

Figure 1e. and Supplementary figure 2d. While these are different blots, the notable difference in phenotype seen between the DMSO control treatments in Figure 1e and Figure 2d is concerning, given that these should recapitulate one another. There is markedly less N-induced degradation of Dicer and other proteins in the supplementary Figure 2d DMSO control compared to Figure 1e, which calls into question the data in Figure 1e.

Response: Thank you for pointing out this issue. These two figures differ due to the discrepancy among experiments conducted in different batches. In the revised manuscript, we quantified the western blots of three biological replicates and selected the appropriate representative images. Please refer to Fig. 1k (Fig. 1e in the previous version) and Supplementary Fig. 2i (Supplementary Fig. 2d in the previous version) in the revised manuscript.

For experiments pointing to autophagy-mediated degradation of these proteins, it would be useful to include additional autophagy inhibitors given their more direct mechanisms of action compared to Chloroquine. This is especially important given that Chloroquine induces extensive disorganization of Golgi and endo-lysosomal systems whereas other available inhibitors are more targeted in their block of lysosome acidification or PI3K complex and early phagosome formation.

Response: We have performed additional experiments using bafilomycin A1, an inhibitor of the late phase of autophagy, blocking the fusion of autophagosomes with lysosomes. The results are presented in Supplementary Fig. 2h in the revised manuscript.

Figure 1f. Including immunofluorescence staining of N protein is important here to show transfection efficiency and link the LC3 phenotype to N protein expression. Including more detailed descriptions on the methodology used for analysis and more information about how many cells were quantified in each experiment would be beneficial as well, as it is unclear what the 200 cell number refers to.

Response: Thank you for these suggestions. We have included immunofluorescence staining of the N protein to show the transfection efficiency and link the LC3 phenotype to N protein expression. The results showed that EGFP-LC3 puncta were formed only in N protein-expressing cells but not in cells not expressing N protein. Please refer to Fig. 2a in the revised manuscript.

As for the quantification methodology, 60–80 cells were counted in each individual biological replicate, and the total cell number of the three biological replicates was 200. We have included this information in the Methods section, “Confocal microscopy,” in the revised manuscript.

Figure 1i. Missing analysis of colocalization between p62 and N—the representative image shown is not convincing and formal analysis of colocalization is needed.

Response: Thank you for this comment. We have quantified the co-localization between p62 and N in cells expressing N protein and cells infected with SARS-CoV-2 using Pearson's correlation coefficient, as previously reported (PMID: 32855409, 23945651, 32513926). The results are presented in Fig. 2c, d in the revised manuscript.

Figure 1h. The presented interaction between N protein and p62 is not robust. Additionally, the IgG control looks as though there is non-specific interactions especially for the Dicer blot.

Response: We apologize for the poor quality of the IP experiment presented in the original manuscript. To eliminate the differences between experiments conducted in different batches, we quantified the western blot experiments of three biological replicates and selected representative images. Please refer to Fig. 2e in the revised manuscript.

Figure 2. Panel A needs quantification to show the phosphorylation induction level of each protein. Blots in Fig S3 need full quantification to show the differences in protein abundance.

Response: We have quantified all western blots in this revision.

The analyses shown in Figs 3 and S4 are compelling but need to be validated in an actual SARS2 infection model, which should be moved from the supplemental figs to these main figs. . Fig 3d-f miRNA nuclear retention assessment lacks important specificity controls to include mRNA and rRNA controls to show specificity of N for retention of miRNA over these other RNAs in the nucleus. rRNA should be included as an internal control for both nuclear and cytosolic RNAs. Figs. 3i-m needs validation of these outcomes in actual SARS2 infected cells-move to main figs.

Response: The analyses shown in Figs 3 and S4 (refer to Figs. 4, 5, and Supplementary Fig. 5 in the revised manuscript) of the original manuscript have been validated using an actual SARS-CoV-2 infection model. The results are presented in Fig. 4c, 4e, 4g, 4i, 4k, 4m, 5c, 5g and 5i in the revised manuscript.

GAPDH and β -actin mRNAs are commonly used as cytosolic internal controls, and U6 snRNA is used as a nuclear internal control (PMID: 32911343, 21297638, 23791529). Therefore, we used GAPDH mRNA and U6 snRNA as internal controls for the biochemical fractionation experiments in this manuscript.

We performed additional experiments to demonstrate that the N protein-induced nuclear retention is specific to pre-miRNAs and does not affect other RNAs. N protein did not induce nuclear retention of 18S RNA and several mRNAs (using GAPDH mRNA and U6 snRNA as internal controls). The data are presented in Fig. 4d, 4e, and Supplementary Fig. 5c in the revised manuscript.

18S rRNA has also been used as an internal control in other papers (PMID: 29567955, 30809316). We conducted new experiments using 18S rRNA as an internal control for both nuclear and cytosolic RNAs. The results are comparable to those obtained when using GAPDH mRNA and U6 snRNA as internal controls. These data are presented below for your evaluation.

Figure for review 4. N protein induced nuclear retention of pre-miRNAs. The ratio of cytosolic levels of different RNAs to their nuclear levels in A549-Ctrl or A549-N cells. 18S rRNA was used as the internal control for both nuclear and cytosolic RNAs.

Figure 4. Unclear about the conclusions regarding protein synthesis v.s protein translation as impacted by N protein given that N protein appears to increase both protein translation and synthesis in the data presented, however description of these same data in the results section contradicts this by stating that N increases translation while inhibiting protein synthesis, which is further contradicted in Supplementary Figure 7. In panel h of Figure 4, reporting densitometry values would be helpful given very subtle differences in blots. Fig 4A needs quantification. This figure shows increased soluble proteins when N is expressed in addition to increased insoluble proteins. This relationship needs to be explained and examined for how N is inducing a general protein upregulation. 4d needs band quantification to show the true differences that the authors are eluding to. Figs 4f-I need quantification.

Response: We apologize for any confusion caused by our previous explanation. The effect of N protein on protein synthesis is two-fold. On one hand, it promotes protein synthesis by reducing the expression of Dicer and XPO5. On the other hand, it inhibits protein synthesis by downregulating the expression of SRSF3 and hnRNPA3. As the downregulation of Dicer and XPO5 has a more significant effect in promoting protein synthesis than the downregulation of SRSF3 and hnRNPA3, N protein ultimately enhances protein synthesis. Moreover, we have quantified all the western blots.

As noted above, the experiments and outcomes in Fig 1-4 all need validation in SARS2 infection models such as in the A549 or Beas2B cells used in these studies. This is a critical omission of the larger study where the N protein regulation of dicer and factors needs to be shown in the context of a real SARS2 infection, and hence the infection data sets should be moved to main figures.

Response: To validate the data obtained with ectopic N protein expression, we performed additional experiments using real SARS-CoV-2 infection models. Please refer to Fig. 1a, 1b, 1f, 1j, 1i, 2d, 2f, 3d, 3e, 3j, 4c, 4e, 4g, 4i, 4k, 4m, 5c, 5g, 5i, 6b, 6e, 6g, 7j, 7k, 9d, 9e, 9f, 10b, 10d, 10e, and Supplementary Fig. 2e, 4f, 7g in the revised manuscript. Furthermore, we have moved the infection data to the main figures.

Fig. 5-7 blots: the bands need to be quantified for protein abundance measurements.

Response: We have quantified all the western blots in this revision.

The experiments in figures 5-7 need to be repeated in mice undergoing SARS2 infection. Since the delta variant SARS2 emergence, the virus is capable of infecting B6 mice, for example or the use a mouse-adapted virus can be applied to do in vivo infection and validation analyses of the fig 5-7 data sets.

Response: To validate the data in Figs 5–7, we infected the hACE2 transgenic C57BL/6J mice with real SARS-CoV-2 virus. Please refer to Fig. 7j, 7k, 9e, 9f, 10d, and 10e, and Supplementary Fig. 7g in the revised manuscript.

Figure 7. The dosage used for anastrozole in mice (20mg/kg) is far higher than what is used in human patients for breast cancer (1 mg/day)—given possible safety concerns about such high doses, it may be important to discuss the viability of using this as a treatment in humans. How old are the mice used in these experiments? It is important to supply evidence that anastrozole reverses the age-related phenotype of worse pneumonia in older individuals beyond just the N-driven changes in younger mice. This would be an especially valuable experiment to include if proposing to use this drug as a treatment for human COVID-19.

Response: We have conducted experiments to test the effects of low anastrozole concentrations on N protein-induced pneumonia. Clinical data have confirmed the safety of long-term administration of low-dose anastrozole (1 mg per day) (PMID: 35123662, 29797697, 34871401). Furthermore, preclinical findings have revealed that short-term administration of high-dose (20 mg/kg) anastrozole is safe (PMID: 32483416). We found that anastrozole dose-dependently ameliorated N protein-induced pneumonia (Fig. 10c). Therefore, further studies are needed to determine the optimal dosage of anastrozole for COVID-19 treatment.

Moreover, we have performed additional experiments to compare the effects of anastrozole on N protein-induced pneumonia in 8-week-old and 18-month-old mice. We found that anastrozole partially rescued the N-induced downregulation of Dicer, XPO5, SRSF3, and hnRNPA3 and ameliorated N protein-induced pneumonia in both groups. The results are presented in Fig. 10f and Supplementary Fig. 10e in the revised manuscript.

Supplementary Figures 1, 2. Having protein-level data to validate the transcript data in figure 1 (a,b) would be valuable, especially given that later in the paper, in Supplementary Figure 2 (panel g) it is stated that SARS-CoV-2 infection did not affect Dicer, XPO5, SRSF3, hnRNPA3 at the mRNA level and infection induces changes only at the protein level. This calls into question what the heatmaps in Supplementary Figure 1a, 1b are showing and if this data is relevant or contradicting the experimental findings of this paper.

Response: We speculate that the decreased mRNA levels of Dicer, XPO5, SRSF3, and hnRNPA3 in the lung tissues of severe and fatal COVID-19 patients are not caused by the SARS-CoV-2 virus directly suppressing these mRNA expressions. Instead, the basal levels of these genes in the lung tissues of these patients were lower than those of patients with mild disease before SARS-CoV-2 infection. Additionally, SARS-CoV-2 infection hinders the expression of these genes at the protein level. Therefore, data in Supplementary Figure 1 (a, b) is consistent with rather than contradicting the data in Supplementary Figure 2 (panel g) (refer to Supplementary Fig. 2e in the revised manuscript), which showed that SARS-CoV-2 infection did not affect Dicer, XPO5, SRSF3, or hnRNPA3 at the mRNA level but decreased their expression at the protein level.

Moreover, as data in supplementary Figures 1 (a, b) were downloaded from the GEO database, acquiring protein-level data to validate the mRNA level using the same batch of cell or tissue samples is impossible. Furthermore, nowadays, as there are very few severe and fatal COVID-19 cases, it is also complicated to obtain lung tissues from patients with severe and fatal COVID-19.

Reviewer #3 (Remarks to the Author):

This manuscript explores the role of the N protein of SARS-CoV-2 in affecting the levels of key proteins of the RNAi pathway and the splicing machinery. The authors show that N expression translates into decreased protein expression, leading to proteotoxic stress and DNA damage. Because the factors targeted by N have decreased levels in aged mice, the authors suggest that N activity towards such factors could contribute to the severity the symptoms in aged individuals. They finish by testing two compounds that could be promising in alleviating N-driven degradation. The manuscript is rather data-heavy, well written and pleasant to read. Appart from data presentation suggestions (detailed after) the experiments are well-controlled and interpreted. It is nice work, but I have a major concern with the experimental setup. The authors perform all their experiments with plasmid-driven N overexpression. During an infection with SARS-CoV-2, the N protein will be loaded on viral RNA, therefore unable to

massively bind cellular RNAs. Because the interaction of N with RNAi and splicing factors depends on RNA, one could expect that it would not be as good of a binder of such cellular factors. The authors need to demonstrate that what they see with plasmid-driven overexpression occurs during viral infection. It would be unrealistic to repeat the entire paper with virus, but key experiments should be performed with actual infection (including in vivo experiments).

Response: Thank you for your positive and constructive comments. We conducted additional experiments to demonstrate that the phenomenon observed with plasmid-driven overexpression also occurred during viral infection. The data are presented in (Fig. 1a, 1b, 1f, 1j, 1i, 2d, 2f, 3d, 3e, 3j, 4c, 4e, 4g, 4i, 4k, 4m, 5c, 5g, 5i, 6b, 6e, 6g, 7j, 7k, 9d, 9e, 9f, 10b, 10d, 10e, and Supplementary Fig. 2e, 4f, 7g) in the revised manuscript.

During viral infection, interaction between viral proteins within virus particles and cellular proteins is frequently observed. Therefore, it is expected that the N protein can interact with cellular proteins in cells infected with SARS-CoV-2. When the virus infects a cell, it must first undergo uncoating before viral RNA replication, transcription, and translation can occur. The N protein may be released from the virus particle during uncoating and then interact with cellular proteins. Moreover, prior to virus assembly, the newly synthesized N protein could potentially bind to cellular proteins.

Major comments

– all the Western blots and microscopy images need to be properly quantified and differences represented as bar graphs. The proper statistical tests need to be performed.

Response: We have quantified all the western blots and microscopy images in the revised manuscript. However, due to space limitation, we displayed the mean values of the protein expression levels from three independent experiments below the western blot instead of using bar graphs.

– what is the control for mouse experiments? The authors should express an unrelated protein and not only infuse mice with a plasmid, to control for the effect of plasmid-driven protein production.

Response: We used viral protein NSP8 as a control and found that ectopic expression of NSP8 in mouse lung tissues did not affect lung injury (Supplementary Fig. 7d).

– If anastrozole acts by enhancing Dicer expression, how come that it also affects XPO5 and splicing factors expression?

Response: Thank you for this comment. As stated above, the molecular mechanisms underlying anastrozole-mediated induction of the expression of Dicer, XPO5, SRSF3, and hnRNPA3

remain to be elucidated. However, this is out of the scope of this study and will be investigated in the future.

Minor comments

– Is the interaction of N with p62 RNA-dependent?

Response: We have performed additional experiments and found that treatment with RNases did not obviously disrupt the interaction between N protein and p62 (Fig. 1h), suggesting that this interaction is RNA-independent.

– gene expression maps: please indicate names of the genes of interest, for example DICER in Suppl Fig 1a. For Fig 3a, please label with the miRNA names.

Response: As suggested, we have indicated the names of the genes and miRNAs in the heatmap; please refer to Supplementary Fig. 1a, b, Fig. 4a, and Supplementary Fig. 5a in the revised manuscript.

REVIEWERS' COMMENTS

Reviewer #1 (Remarks to the Author):

I am ok with the revised version of this ms

Reviewer #2 (Remarks to the Author):

The authors have addressed concerns and the presentation is improved and conclusive.

Reviewer #3 (Remarks to the Author):

The authors performed a lot of work to tackle the revisions and answered all my questions and comments. I therefore recommend the manuscript for publication.